# Assessing the influence of lake and watershed attributes on snowmelt bypass at thermokarst lakes

Evan J. Wilcox[1], Brent B. Wolfe[1], and Philip Marsh[1]

[1]Department of Geography and Environmental Studies, Wilfrid Laurier University, Waterloo, N2L3C5, Canada

**Correspondence:** E. J. Wilcox (evan.j.wilcox@gmail.com)

**Abstract.** Snow represents the largest potential source of water for thermokarst lakes, but the runoff generated by snowmelt (freshet) can flow beneath lake ice and via the outlet without mixing with and replacing pre-snowmelt lake water. Although this phenomenon, called "snowmelt bypass", is common in ice-covered lakes, it is unknown what lake and watershed properties cause variation in snowmelt bypass among lakes. Understanding the variability of snowmelt bypass is important because the amount of freshet that is mixed into a lake affects the hydrological and biogeochemical properties of the lake. To explore lake and watershed attributes that influence snowmelt bypass, we sampled 17 open-drainage thermokarst lakes for isotope analysis before and after snowmelt. Isotope data were used to estimate the amount of lake water replaced by freshet and to observe how the water source of lakes changed in response to the freshet. Among the lakes, a median of 25.2% of lake water was replaced by freshet, with values ranging widely from 5.2 to 52.8%. For every metre lake depth increased, the portion of lake water replaced by freshet decreased by an average of 13%, regardless of the size of the lake's watershed. The thickness of the freshet layer was not proportional to maximum lake depth, so that a relatively larger portion of pre-snowmelt lake water remained isolated in deeper lakes. We expect a similar relationship between increasing lake depth and greater snowmelt bypass could be present at all ice-covered open-drainage lakes that are partially mixed during the freshet. The water source of freshet that was mixed into lakes was not exclusively snowmelt, but a combination of snowmelt mixed with rain-sourced water that was released as the soil thawed after snowmelt. As climate warming increases rainfall and shrubification causes earlier snowmelt timing relative to lake ice melt, snowmelt bypass may become more prevalent with the water remaining in thermokarst lakes post-freshet becoming increasingly rainfall sourced. However, if climate change causes lake levels to fall below the outlet level (i.e., lakes become closed-drainage), more freshet may be retained by thermokarst lakes as snowmelt bypass will not be able to occur until lakes reach their outlet level.

## 1 Introduction

In the continuous permafrost zone of the Arctic, regions with thermokarst lakes have formed where ice-rich permafrost has thawed and the ground surface has subsided. Thermokarst lakes typically range from 1 – 5 m in depth, 0.01 – 1000 ha in area, can cover over 25% of the land area (Grosse et al., 2008; Burn and Kokelj, 2009; Turner et al., 2014; Farquharson et al., 2016) and mostly formed during a brief warm period following the last deglaciation of the northern hemisphere (Brosius et al., 2021). Comparison of aerial photography from the mid-1900s with more recent satellite imagery has revealed both increases

and decreases in thermokarst lake area and number (Smith et al., 2005; Plug et al., 2008; Marsh et al., 2009; Jones et al., 2011; Finger Higgens et al., 2019). These changes are partially attributed to shifting thermokarst lake water balances: increased air temperatures (Woo et al., 2008), longer ice-free seasons (Surdu et al., 2014; Arp et al., 2015), permafrost thaw (Walvoord and Kurylyk, 2016), and shrub expansion leading to increased transpiration (Myers-Smith et al., 2011) and interception (Zwieback

et al., 2019), all cause less inflow and more water to evaporate from thermokarst lakes. Contrarily, increasing precipitation can lead to more inflow to lakes, offsetting any rise in evaporation, interception and transpiration (Walsh et al., 2011; Stuefer et al., 2017; Box et al., 2019; MacDonald et al., 2021), while shrub expansion can also increase snow accumulation in lake watersheds resulting in more snowmelt runoff to lakes (Turner et al., 2014; MacDonald et al., 2017). Increased rainfall has also been linked to decreases in lake surface area because lakes are more likely to experience rapid drainage due to permafrost thaw

during wet years (Webb et al., 2022).

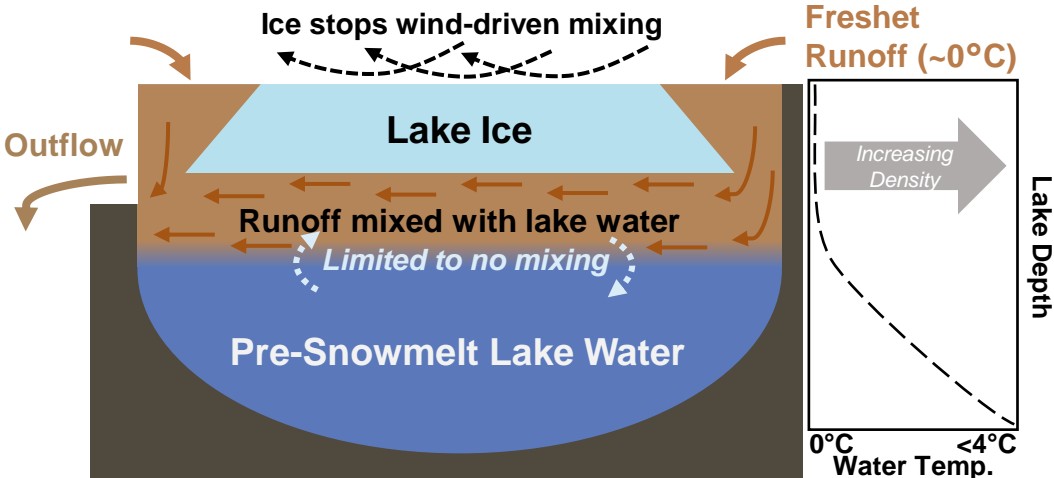

**Figure 1.** A conceptual cross-section of an open-drainage lake when freshet has begun. Freshet initially flows into the lake at the edge where lake ice has melted. A layer of snowmelt runoff mixed with lake water then remains buoyant on top of the warmer lake water, before flowing through the outlet (i.e., 'snowmelt bypass'). Limited mixing occurs due to density differences between runoff and deeper lake water, and the lack of wind-driven mixing due to the presence of lake ice.

Runoff generated by snowmelt in lake watersheds represents a large potential water source for lakes, as snowfall comprises 40 to 80% of total precipitation in the Arctic (Bintanja and Andry, 2017). When snow melts in spring, the volume of snowmelt-driven runoff flowing into lakes (freshet) generally results in the highest lake levels of the year (Woo, 1980; Roulet and Woo, 1988; Hardy, 1996; Pohl et al., 2009). When freshet is low, thermokarst lakes are prone to desiccation (Marsh and Bigras, 1988;

Marsh and Lesack, 1996; Bouchard et al., 2013). It is a reasonable expectation that lakes which receive more freshet will also contain more freshet by the end of the snowmelt if they remain below their outlet level (i.e., closed-drainage lakes). However, for non-bedfast ice-covered lakes at or near their outlet level (i.e., open-drainage lakes), freshet may flow into and out of a lake without mixing with and replacing the pre-freshet lake water, resulting in "snowmelt bypass" (Bergmann and Welch, 1985)

(Figure 1). While lake ice inhibits wind-driven mixing of lake water, the cooler, less dense freshet (~0 °C) cannot mix with the deeper, warmer, and denser lake waters (<4 °C). As a result, freshet water will flow into and out of an open-drainage lake without replacing the deeper, pre-snowmelt lake water until vertical mixing within the lake begins, which is initiated by the warming of lake waters from solar radiation penetrating through snow-free ice, and wind-driven mixing after the lake becomes ice-free (Cortés and MacIntyre, 2020). Snowmelt bypass is a common occurrence that has been observed in a wide variety of ice-covered open-drainage lakes around the world (Henriksen and Wright, 1977; Jeffries et al., 1979; Hendrey et al., 1980; Bergmann and Welch, 1985; Schiff and English, 1988; Edwards and McAndrews, 1989; Cortés et al., 2017).

Although previous studies have established the mechanisms and conditions that cause snowmelt bypass, no studies have examined how lake and watershed characteristics affect snowmelt bypass. Given that snowmelt bypass depends on the mixing conditions under lake ice, we hypothesize that lake and watershed characteristics that impact lake mixing may cause variability in snowmelt bypass among lakes in a given region. Understanding the factors that influence the amount of freshet retained by thermokarst lakes is important because of subsequent influence on lake water balance, pH, nutrient composition, and suspended sediment, among other limnological variables (Henriksen and Wright, 1977; Marsh and Pomeroy, 1999; Finlay et al., 2006; Turner et al., 2014; Balasubramaniam et al., 2015).

In this study, we determine factors influencing the magnitude of snowmelt bypass for 17 open-drainage thermokarst lakes in the lake-rich tundra uplands east of the Mackenzie Delta in the Northwest Territories, Canada, during the freshet of 2018. This area contains thousands of thermokarst lakes that constitute up to 25% of the landscape surface area and have changed in area and number during the past several decades in response to changing precipitation and permafrost thaw (Plug et al., 2008; Marsh et al., 2009). We used isotopic composition of lake water and precipitation from before and after snowmelt to estimate the proportion of lake water replaced by freshet during spring 2018 and evaluated relations with lake and watershed characteristics. We selected lake and watershed characteristics that had potential to impact under-ice mixing through their influence on the water temperature profile (e.g. lake depth) or by displacing pre-snowmelt lake water (e.g. watershed area). Isotope tracers were also used to assess whether the freshet is sourced solely from snowmelt, or if other water sources contributed to freshet. Future assessments of hydrological and biogeochemical properties of thermokarst lakes can use the lake and watershed attributes we identify to affect snowmelt bypass and lake water sources to inform their results, given the distinct biogeochemical properties of freshet runoff (Finlay et al., 2006; Balasubramaniam et al., 2015) and the influence of snowmelt bypass on the amount of freshet runoff retained by lakes.

## 2 Study Area

The 17 study lakes are situated in the taiga-tundra uplands east of the Mackenzie Delta, in the northwest region of the Northwest Territories, Canada (Figure 2). The landscape is comprised of rolling hills and is strongly influenced by permafrost thaw, as evidenced by the thousands of thermokarst lakes which formed between 13 000 and 8000 years ago (Rampton, 1988; Burn and Kokelj, 2009) that are typically 2 – 4 m in depth with a surface area from 10 – 1000 ha (Pienitz et al., 1997). The study lakes are situated along a ~70 km stretch of the Inuvik-Tuktoyaktuk Highway north of the town of Inuvik (Figure 2). The average

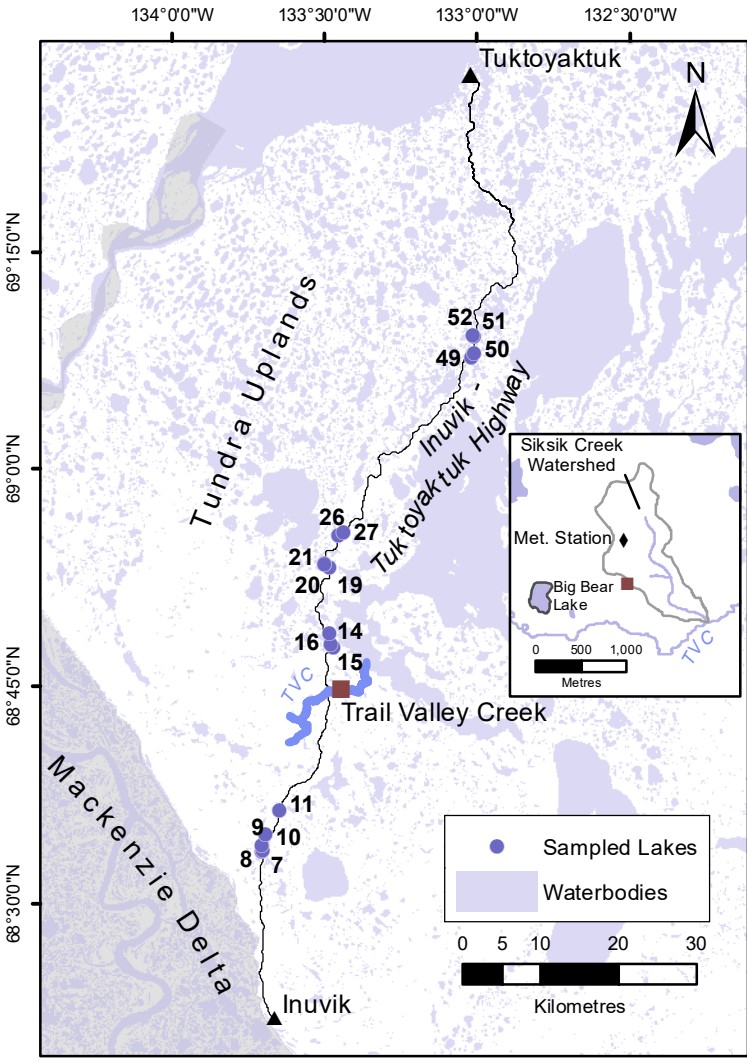

**Figure 2.** Blue circles indicate lakes that were sampled before and after snowmelt in 2018. Tundra uplands are in white while the Mackenzie Delta is in grey. In the inset, key locations near the Trail Valley Creek field station are shown.

area of the lakes is 14.2 ha (0.9 – 90.5 ha) and the average maximum depth is 2.2 m (1.0 – 4.1 m) (Table 1). All lakes have a defined outlet channel observed to be active during the spring melt, thus classifying them hydrologically as open-drainage, and many lakes have defined channelized inflows from their watersheds in the form of small streams or ice-wedge polygon troughs.

Soils in the region have evolved from fine-grained morainal tills, ice-contact sediment, and lacustrine deposits (Rampton and Wecke, 1987). Subsurface flow is conveyed efficiently by a network of interconnected peat channels 0.3 – 1.0 m wide that lay between mineral earth hummocks (Quinton and Marsh, 1998). Lake watersheds contain tall shrub (>1 m), low-shrub (~0.5 m), and shrub-free landcover types comprising lichen, moss, and tussocks (Lantz et al., 2010; Grünberg et al., 2020). Mean annual air temperature in Inuvik is -8.2°C and mean annual precipitation is 241 mm of which 66% is snow, based on 1981-2010 climate normals (Environment and Climate Change Canada, 2019). Snowmelt usually begins in mid-May, and lakes typically become ice-free in June and freeze-up in mid-October (Burn and Kokelj, 2009).

The 2018 snowmelt season was typical in comparison to recent decades. End of winter snow surveys conducted in the 58 km$^2$ watershed of Trail Valley Creek in 2018 (Figure 2) recorded an average snow water equivalent (SWE) of 141 mm, close to the average SWE of 147±35 mm based on surveys from 1991 – 2019 (Marsh et al., 2019). At Trail Valley Creek, snowmelt began about May 1, with snow-free areas beginning to appear by May 8, while only the remnants of large snow drifts remained by June 3. Lake ice near Trail Valley Creek became snow-free by May 10, and lakes became completely ice-free on June 14. The mean air temperature at Trail Valley Creek during the sampling period from April 26 – June 15, 2018 was $0.4\,°C$, which was cooler than the average of $1.7\,°C$ during 1999-2019 (Figure 3). Air temperatures roughly followed the average minimum and maximum daily air temperatures, with some temporal variability which can be expected for any given year. Maximum daily air temperatures were mostly above $0\,°C$ after May 8, which was similar to the average timing of the first above $0\,°C$ day during 1999-2019 (Figure 3).

## 3 Methods

### 3.1 Lake water and precipitation sampling for isotope analysis

Lake water samples for stable isotope analysis were first collected from the 17 study lakes while they were ice covered (April 26 – May 1, 2018) and again soon after lakes became ice-free (June 15, 2018). Pre-snowmelt water samples were obtained from a hole augured through the ice near the centre of each lake. These water samples were taken 10 cm below the water surface in the augured hole. Lake depth, snow depth on the ice, and ice thickness were recorded at the same time water samples were collected. Snow depth was typically very uniform across the lake ice. Water samples were then collected post-snowmelt at the shore of each lake shortly after the lakes became fully ice-free. Isotope data were then used to estimate the portion of lake water that was replaced by freshet between the two sampling dates.

To estimate the Local Meteoric Water Line (LMWL) and the average isotopic composition of precipitation ($\delta_P$) in the study region, which are useful references for the interpretation of lake water isotopic compositions, samples of end-of-winter snow on the ground in April 2018 and rainfall for the period May to September 2018 were obtained. Snow samples (n = 11) were

**Table 1.** Lake and watershed properties for sampled lakes. Lake locations are shown on Figure 2

| Lake | Longitude | Latitude | Lake Elevation (m asl) | Lake Depth (m) | Ice Thickness (m) | Snow Depth (cm) | Lake Area (ha) | Watershed Area (ha) | Watershed Area/Lake Area |
|---|---|---|---|---|---|---|---|---|---|
| 7 | -133.76149 | 68.55745 | 89 | 2.24 | 0.81 | 22 | 2.81 | 6.45 | 2.62 |
| 8 | -133.75566 | 68.55879 | 89 | 2.30 | 0.79 | 15 | 1.88 | 15.67 | 7.55 |
| 9 | -133.76025 | 68.56446 | 86 | 1.02 | 0.84 | 30 | 59.56 | 203.56 | 3.42 |
| 10 | -133.74651 | 68.57601 | 88 | 1.65 | 0.85 | 54 | 90.48 | 168.58 | 1.86 |
| 11 | -133.70334 | 68.60390 | 83 | 1.91 | 0.97 | 11 | 0.92 | 21.76 | 17.32 |
| 14 | -133.52093 | 68.78877 | 52 | 1.42 | 0.84 | 4 | 10.68 | 60.64 | 5.7 |
| 15 | -133.52885 | 68.79452 | 57 | 1.57 | 1.14 | 19 | 5.66 | 29.83 | 4.99 |
| 16 | -133.53196 | 68.80550 | 52 | 3.18 | 1.32 | 7 | 1.15 | 19.75 | 16.02 |
| 19 | -133.52616 | 68.88175 | 39 | 2.46 | 1.24 | 11 | 5.68 | 38.98 | 7.09 |
| 20 | -133.54301 | 68.88474 | 37 | 2.69 | 1.27 | 10 | 2.30 | 19.93 | 9.18 |
| 21 | -133.54002 | 68.88721 | 36 | 1.78 | 1.19 | 11 | 2.61 | 10.91 | 3.96 |
| 26 | -133.49557 | 68.91814 | 38 | 1.47 | 1.19 | 6 | 4.84 | 17.89 | 3.83 |
| 27 | -133.47711 | 68.92095 | 45 | 3.10 | 1.22 | 10 | 1.13 | 8.57 | 6.7 |
| 49 | -133.05281 | 69.11883 | 9 | 2.18 | 0.91 | 23 | 17.50 | 46.23 | 2.54 |
| 50 | -133.04203 | 69.12333 | 8 | 1.65 | 0.86 | 19 | 8.16 | 31.92 | 3.67 |
| 51 | -133.04142 | 69.14222 | 4 | 2.31 | 0.84 | 24 | 2.24 | 12.01 | 5.26 |
| 52 | -133.04730 | 69.14389 | 6 | 4.14 | 0.86 | 18 | 23.52 | 49.92 | 2.05 |
| Min | -133.76149 | 68.55745 | 4 | 1.02 | 0.79 | 4 | 0.92 | 6.45 | 1.86 |
| Mean | -133.47497 | 68.83944 | 48 | 2.18 | 1.01 | 17 | 14.18 | 44.86 | 6.10 |
| Max | -133.04142 | 69.14389 | 89 | 4.14 | 1.32 | 54 | 90.48 | 203.56 | 17.32 |

collected from the study area by taking a vertical core of snow using a tube, completely melting the snow in a sealed plastic bag, and then filling a sample bottle with the meltwater. Rainfall (n = 13) was collected between May and September in Inuvik using a clean high-density polyethylene container, which was then transferred to a sample bottle shortly after the rain had stopped. The midpoint between the average isotopic composition of snow samples and rain samples was used to define $\delta_P$. All samples were collected in 30 mL high-density polyethylene bottles and were measured using an isotope analyzer to determine the ratio of $^{18}O/^{16}O$ and $^2H/^1H$ in each sample. Isotope concentrations were measured using a Los Gatos Research (LGR) Liquid Water Isotope Analyser, model T-LWIA-45-EP at the Environmental Isotope Laboratory at the University of Waterloo. The instrument was calibrated using Vienna Mean Standard Ocean Water (VSMOW) and Vienna Standard Light Antarctic Precipitation (VSLAP) standards provided by LGR. Calibration of the instrument was checked during the analysis using the VSMOW and VSLAP standards. Isotopic compositions are expressed in standard $\delta$-notation, such that:

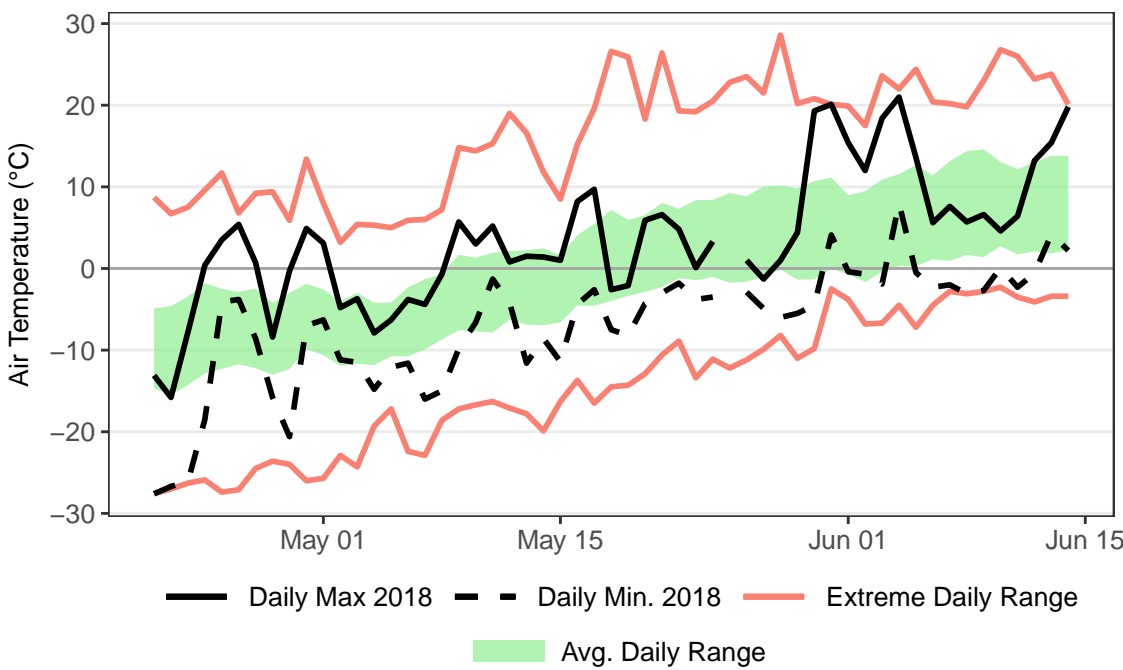

**Figure 3.** Maximum and minimum daily air temperature in comparison to the average and extreme values at the Trail Valley Creek field station for the period of 1999-2019.

$$\delta_{sample} = \frac{R_{sample}}{R_{VSMOW}} - 1 \tag{1}$$

where R represents the ratio of $^{18}O/^{16}O$ or $^2H/^1H$. Every fifth sample was analyzed a second time to determine the analytical uncertainties, which were ±0.1‰ for $\delta^{18}O$ and ±0.6‰ for $\delta^2H$, calculated as two standard deviations away from the difference between the duplicate samples. All isotope data from lakes are presented in Table 2.

### 3.2 Estimating the replacement of lake water by freshet and lake source waters

The percentage of a lake's volume that has been replaced by a given water source can be estimated as follows:

$$\% \, lake \, water \, replaced = \frac{\delta_{L-Post} - \delta_{L-Pre}}{\delta_{I-Post} - \delta_{L-Pre}} * 100 \tag{2}$$

where $\delta_{L\text{-Pre}}$ is the isotopic composition of lake water before snowmelt begins, $\delta_{L\text{-Post}}$ is the isotopic composition of lake water after snowmelt is complete, and $\delta_{I\text{-Post}}$ is the isotopic composition of the source water post-snowmelt. Application of this equation assumes minimal to no change in volume, which is reasonable given the lakes we sampled are all open-drainage. We

also assume that lakes are well mixed as they become ice-free when $\delta_{L\text{-Post}}$ water samples were collected. Water temperature measurements at Big Bear Lake (Figure 2), previous water temperature profiles made at lakes within 10 km of the Inuvik-

**Table 2.** Snow and ice thickness, isotope, $\delta_I$, and lake water replacement values for all lakes. Ice-Corrected $\delta_L$ values were used to calculate Ice-Corrected $\delta_I$ values following Yi et al. (2008).

| Lake | Ice Thickness (m) | Snow Depth (cm) | Pre-Snowmelt Lake Water (26-04-2018 to 01-05-2018) $\delta^{18}O$ | $\delta^2H$ | $\delta^{18}O$ Ice-Corrected | $\delta^2H$ Ice-Corrected | $\delta^{18}O_I$ Ice-Corrected | $\delta^2H_I$ Ice-Corrected | Post-Snowmelt Lake Water (15-06-2018) $\delta^{18}O$ | $\delta^2H$ | $\delta^{18}O_I$ | $\delta^2H_I$ | Average % lake water replaced |
|---|---|---|---|---|---|---|---|---|---|---|---|---|---|
| 7 | 2.24 | 0.81 | -15.41 | -131.05 | -14.07 | -123.16 | -22.84 | -171.44 | -15.48 | -129.67 | -21.11 | -159.22 | 19.1 |
| 8 | 2.30 | 0.79 | -15.94 | -134.71 | -14.7 | -127.42 | -24.68 | -184.42 | -16.71 | -137.71 | -22.9 | -171.81 | 23.9 |
| 9 | 1.02 | 0.84 | -17.17 | -140.98 | -12.02 | -110.51 | -17.94 | -136.81 | -15.45 | -128.25 | -19.78 | -149.79 | 44.7 |
| 10 | 1.65 | 0.85 | -15.68 | -132.02 | -13.54 | -119.35 | -20.84 | -157.31 | -14.86 | -125.77 | -20.29 | -153.37 | 19.3 |
| 11 | 1.91 | 0.97 | -20.56 | -156.48 | -18.48 | -144.48 | -20.47 | -154.65 | -19 | -147.33 | -20.59 | -155.52 | 25.2 |
| 14 | 1.42 | 0.84 | -19.85 | -154.24 | -17.23 | -139.05 | -21.3 | -160.52 | -18.66 | -146.31 | -21.09 | -159.04 | 36.7 |
| 15 | 1.57 | 1.14 | -18.17 | -144.83 | -14.35 | -122.41 | -19.35 | -146.76 | -16.03 | -132.27 | -20.82 | -157.11 | 27.2 |
| 16 | 3.18 | 1.32 | -20.57 | -155.32 | -18.99 | -146.18 | -19.96 | -151.05 | -19.05 | -146.66 | -20.05 | -151.69 | 7.2 |
| 19 | 2.46 | 1.24 | -18.92 | -149.63 | -16.85 | -137.58 | -21.86 | -164.46 | -18.26 | -144.16 | -21.05 | -158.75 | 32.4 |
| 20 | 2.69 | 1.27 | -19.32 | -149.44 | -17.44 | -138.52 | -20.02 | -151.49 | -17.86 | -140.53 | -19.96 | -151.07 | 16.3 |
| 21 | 1.78 | 1.19 | -19.61 | -154.6 | -16.33 | -135.57 | -22.83 | -171.31 | -17.72 | -141.63 | -21.29 | -160.47 | 26.2 |
| 26 | 1.47 | 1.19 | -17.5 | -141.72 | -12.59 | -112.7 | -17.47 | -133.48 | -16.15 | -131.84 | -19.87 | -150.41 | 49.9 |
| 27 | 3.10 | 1.22 | -17.95 | -144.85 | -16.48 | -136.26 | -22.68 | -170.25 | -17.79 | -142.41 | -21.65 | -162.98 | 24.2 |
| 49 | 2.18 | 0.91 | -14.72 | -124.46 | -13.12 | -114.91 | -17.44 | -133.22 | -14.75 | -123.4 | -18.51 | -140.81 | 31.6 |
| 50 | 1.65 | 0.86 | -14.99 | -127.6 | -12.8 | -114.59 | -18.86 | -143.31 | -15.43 | -125.22 | -17.7 | -135.07 | 52.8 |
| 51 | 2.31 | 0.84 | -16.28 | -133.15 | -14.95 | -125.31 | -19.33 | -146.63 | -15.8 | -129.92 | -19.72 | -149.36 | 18.5 |
| 52 | 4.14 | 0.86 | -13.98 | -120.75 | -13.29 | -116.63 | -18.51 | -140.79 | -13.59 | -117.58 | -18.03 | -137.45 | 5.5 |
| Min | 1.02 | 0.79 | -20.57 | -156.48 | -18.99 | -146.18 | -24.68 | -184.42 | -19.05 | -147.33 | -22.9 | -171.81 | 5.2 |
| Mean | 2.18 | 1.01 | -17.45 | -140.93 | -15.13 | -127.33 | -20.38 | -154.00 | -16.62 | -134.74 | -20.26 | -153.17 | 27.1 |
| Max | 4.14 | 1.32 | -13.98 | -120.75 | -12.02 | -110.51 | -17.44 | -133.22 | -13.59 | -117.58 | -17.7 | -135.07 | 52.8 |

Tuktoyaktuk Highway, and lake temperature modelling using FLake-online (Kirillin et al., 2011) all suggest that lakes were well mixed at the time of $\delta_{L\text{-Post}}$ sampling on 2018-06-15 (Appendix A).

We calculated $\delta_I$ following the coupled isotope tracer approach outlined by Yi et al. (2008), using an isotope framework based on 2017 air temperature and humidity data for the typical ice-free period (June 15 – October 15) collected at the Trail Valley Creek meteorological station located 45 km NNE of Inuvik (Figure 2). Data from 2017 were used because it was the most recent period where lakes were exposed to meteorological conditions for an entire open-water season. The coupled isotope tracer approach assumes all lakes under the same meteorological conditions will evolve towards the same isotopic composition ($\delta^*$, the isotopic composition of a lake at the moment of desiccation) as lakes evaporate along lake-specific evaporation lines. These lake-specific evaporation lines are defined by extrapolating from $\delta^*$ through $\delta_L$ until intersection with the Local Meteoric Water Line, which is used to estimate $\delta_I$ (Figure 4). We calculated $\delta_I$ for pre-snowmelt and post-snowmelt isotopic composition of lake water to identify whether the isotopic composition of the source water changed after freshet. The percentage of lake water replaced was calculated using both $\delta^{18}O$ and $\delta^2H$ using Equation 2 and average values are reported. The difference obtained using the two isotopes in the estimate of the percentage of lake volume replaced by runoff was minimal, averaging 1.8% and ranging from 0.6 – 3.7%.

A sensitivity analysis was performed to evaluate uncertainty in the $\delta_I$ values and subsequent % lake water replaced by runoff. Confidence in the interpretation of $\delta_I$ values with respect to rainfall- or snowmelt-sourced waters depends on an accurate estimate of $\delta_P$, which was determined using the average $\delta_{Rain}$ and $\delta_{Snow}$ values. Also, the calculation of % lake water replaced by runoff is sensitive to changes in $\delta_I$, which is sensitive to the average $\delta_{Rain}$ value because this parameter is used to determine $\delta_{As}$ (Equation B5), which is subsequently used to determine $\delta^*$ (Equation B1). Since there was some variability in the $\delta_{Rain}$ and $\delta_{Snow}$ values from samples we collected, we tested the sensitivity of our estimates of $\delta_I$ and the percentage of lake water replaced by runoff to variation in average $\delta_{Rain}$ and $\delta_{Snow}$ values. We calculated the probable upper bound and lower bound limits of $\delta_{Rain}$ and $\delta_{Snow}$ values by adding and subtracting the standard error from the average of $\delta_{Rain}$ and $\delta_{Snow}$ values (Appendix B). Upper and lower bound cases were propagated through the isotope framework to calculate upper and lower bound $\delta_I$ and % lake water replacement values to evaluate whether the standard error caused enough deviation to meaningfully change the results. Overall, differences between upper and lower bound $\delta_I$ and % lake water replacement values were minimal (Table B2, Figure B1). Details of the equations and variables used in the isotope framework and the sensitivity analysis are given in Appendix B.

As ice forms and preferentially incorporates water containing the heavy isotopes $^{18}O$ or $^2H$, the lake water beneath the ice becomes increasingly depleted in $^{18}O$ and $^2H$. Consequently, the water samples we collected pre-snowmelt were systematically isotopically depleted relative to pre-freeze-up lake water, and the magnitude of depletion depends on the fraction of lake water that had frozen into ice. We corrected $\delta_{L\text{-Pre}}$ for the fractionation of freezing water into ice using an equation developed by Gibson and Prowse (2002) that describes the fractionation of isotopes between water and freezing ice in a closed system:

$$\delta_{L-Pre} = -f^{\alpha_{eff}}(1000 * f^{\alpha_{eff}} - f * \delta_{L-BelowIce} - 1000 * f) \tag{3}$$

where $\delta_{L\text{-BelowIce}}$ is the isotopic composition of the water beneath the lake ice, $\alpha_{eff}$ is the effective fractionation factor between ice and water, defined as $\alpha_{eff} = R_{Ice}/R_L$, and $f$ is the fraction of unfrozen water remaining in the lake. $\alpha_{eff}$ is dependent on the

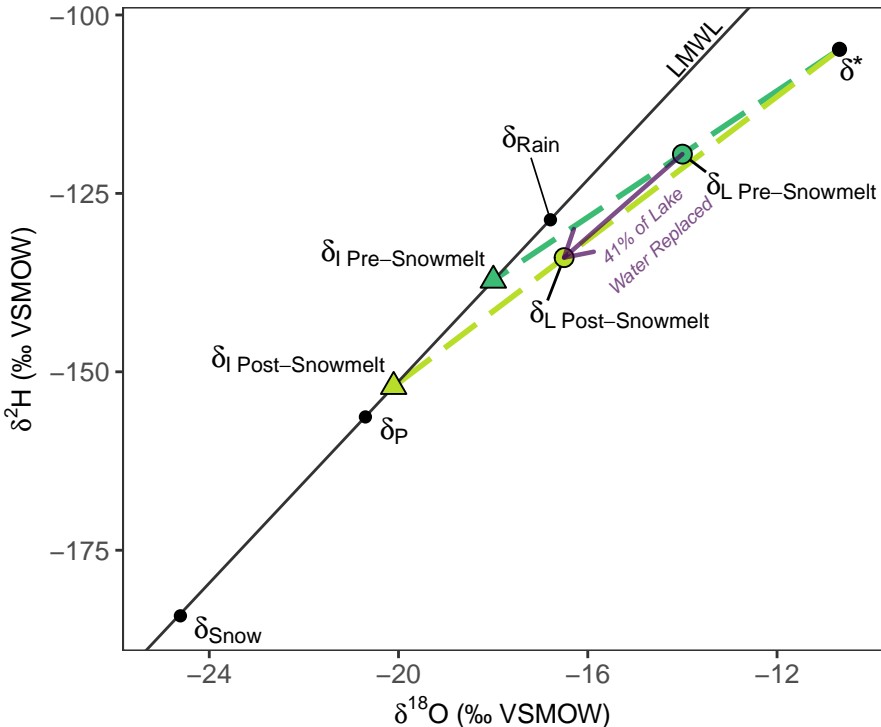

**Figure 4.** A hypothetical change in the isotopic composition of lake water from pre-snowmelt to post-snowmelt is shown. A visualization of how $\delta_I$ is calculated for an individual lake using a lake-specific evaporation line for both pre-snowmelt and post-snowmelt is also shown, where each lake's evaporation line (dashed line) extrapolates from $\delta^*$ through $\delta_L$ until intersecting the Local Meteoric Water Line to give $\delta_I$. The Local Evaporation Line (LEL) is defined by the line between $\delta_P$ and $\delta^*$ (not shown).

thickness of the boundary layer between the forming ice and freezing water and the downwards freezing velocity of the ice. Since we did not have measurements of either of these parameters, we relied on previously estimated values of $\alpha_{eff}$ (Souchez et al., 1987; Bowser and Gat, 1995; Ferrick et al., 2002) and boundary layer thickness (Ferrick et al., 1998, 2002; Gibson and Prowse, 2002). Using this information, we estimated values of $\alpha_{eff}$ that produced $\delta_{L\text{-Pre}}$ values that closely match lake water

isotopic compositions measured at the same lakes in August and September of 2018 (Figure C1). Additional information about the determination of $\alpha_{eff}$ values is provided in Appendix C.

    To estimate the fraction of unfrozen water remaining in lakes ($f$, Equation 3), bathymetry was collected at Big Bear Lake (Figure 2), a typical bowl-shaped thermokarst lake near the Trail Valley Creek meteorological station in June 2017 using a Garmin echoMAP CHIRP 42dv fish finder. Bathymetry data were used to determine a relationship between lake volume and

lake depth. We fit a quadratic equation to the bathymetric data to estimate the fraction of lake volume relative to the fraction of lake depth. The best fit quadratic equation ($r^2 = 0.9997$) was:

$$\%_{LakeVolume} = (-0.0115 * \%_{LakeDepth})^2 + 2.1508 * \%_{LakeDepth} - 0.4857 \tag{4}$$

where $\%_{\text{LakeVolume}}$ is the fraction of total lake volume and $\%_{\text{LakeDepth}}$ is the fraction of total lake depth. However, this fitted equation does not reach $100\%_{\text{LakeVolume}}$ at $100\%_{\text{LakeDepth}}$, or $0\%_{\text{LakeVolume}}$ at $0\%_{\text{LakeDepth}}$, which is required to realistically
represent the relationship between lake depth and lake volume. The equation was slightly adjusted to:

$$\%_{LakeVolume} = (-0.01 * \%_{LakeDepth})^2 + 2 * \%_{LakeDepth} \tag{5}$$

in order to satisfy these requirements, resulting in a mean offset of 1.7% between the measured bathymetric data and the adjusted equation. Most lakes in this region have a bowl-shaped bathymetry because they were formed by thermokarst processes (Rampton, 1988; Burn and Kokelj, 2009), where subsidence caused by the thaw of ice-rich permafrost results in a waterbody
which then expands outward radially in all directions. Bathymetric data for Big Bear Lake and a comparison of the equation between lake volume and depth are provided in Appendix D.

### 3.3   Quantifying lake and watershed properties

We quantified multiple lake and watershed properties to explore relations with the amount of lake water replaced by freshet. These properties included lake depth, lake volume, snow depth on the lake, ice thickness, lake area and watershed area. Lake
depth, snow depth on the lake and ice thickness were measured at the same time as pre-snowmelt lake samples were collected. Lake volume was approximated by multiplying the product of lake depth and lake area by 0.7, a relation derived from the measured lake volume of Big Bear Lake. Watershed area was estimated by applying the D8 water routing algorithm (O'Callaghan and Mark, 1984) to the 2 metre resolution ArcticDEM (PGC, 2018) using ArcGIS 10.7.1 (ESRI, 2019).

### 4   Results

Correcting for ice fractionation using Equation 3 resulted in an increase in estimated $\delta_{\text{L-Pre}}$ values as expected, with the median shifting from -17.50‰ (-19.32‰ to -15.68‰ IQR, inter-quartile range) to -14.70‰ (-16.85‰ to -13.29‰ IQR) for $\delta^{18}$O (Figure 5a, Table 2). The corrected pre-snowmelt isotopic compositions of lake water were distributed across a large range of the predicted LEL, spanning from near the LMWL to near $\delta^*$ (Figure 5a), reflecting that the lake waters were variably influenced by evaporation. Corrected pre-snowmelt isotopic compositions of lake waters also tightly cluster along the LEL,
indicating that the predicted LEL is well characterized.

     The change in the isotopic composition of lake water from pre-snowmelt to post-snowmelt was characterized by a small (~1.5‰ in $\delta^{18}$O) shift towards $\delta_P$, with median pre-snowmelt $\delta_{\text{L-Pre}}$ values of -14.70‰ (-16.85‰ to -13.29‰ IQR) and median $\delta_{\text{L-Post}}$ values of -16.15‰ (-17.86‰ to -15.45‰ IQR) for $\delta^{18}$O (Figure 5b). The small change in the isotopic composition of lake water meant that most lakes retained an evaporated isotope signature post-snowmelt, overlapping with a substantial
portion of $\delta_{\text{L-Pre}}$ and continuing to plot along the LEL (Figure 5b). Post-snowmelt, about half of the lakes (9 of 17) also plotted above the LEL, indicating that the $\delta_I$ of these lakes was more similar to rainfall than snowfall (Figure 5b). The shift in $\delta_I$ for lakes from pre-snowmelt to post-snowmelt shows a convergence of most $\delta_I$ values towards a value near $\delta_P$ and away from the isotopic composition of the end-of-winter snow ($\delta_{\text{Snow}}$) or rainfall ($\delta_{\text{Rain}}$) (Figure 5c). The convergence of $\delta_I$ values towards $\delta_P$

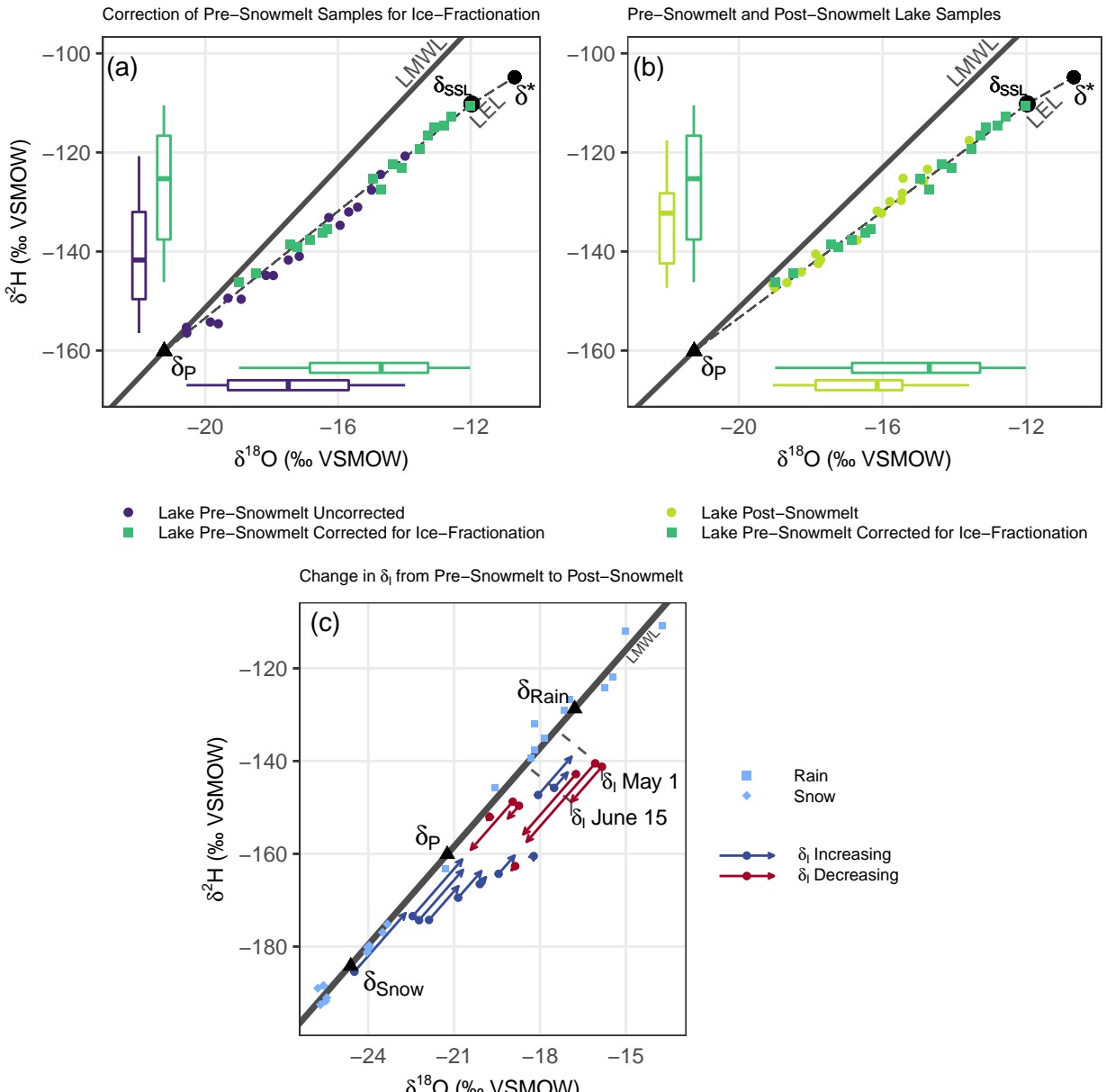

**Figure 5.** Lake water and precipitation isotope data are displayed on $\delta^{18}O$-$\delta^2H$ graphs. The Local Meteoric Water Line (LMWL: $\delta^2H = 7.1*\delta^{18}O - 10.0$) is indicated by the solid line, while the Local Evaporation Line (LEL: $\delta^2H = 5.2*\delta^{18}O - 48.9$) is indicated by the dashed line. $\delta_P$ represents the average value of precipitation in the region, based on 2018 sampling of end-of-winter snow and rainfall from April to September. $\delta_{SSL}$ is the point at which evaporation and inflow are equal (E/I = 1). (a) Uncorrected and corrected for ice fractionation pre-snowmelt lake water isotope data. (b) Corrected pre-snowmelt and post-snowmelt data. (c) The shift in $\delta_I$ from pre-snowmelt to post-snowmelt, as indicated by a circle for pre-snowmelt $\delta_I$ values and the end of the arrow for post-snowmelt $\delta_I$ values. All $\delta_I$ values are offset from the LMWL for visibility, as all $\delta_I$ values are constrained to the LMWL.

and away from end-of-winter snow signify that a non-snow source of water, with a more enriched isotopic composition than

$\delta_{\text{Snow}}$, was present in freshet.

Replacement of lake water by freshet ranged widely from 5.2 – 52.8%, with a median of 25.2% (19.1% to 32.4% IQR, Figure 6). A substantial proportion of this variation was explained by lake depth: deeper lakes had significantly less of their water replaced by freshet, with a reduction in lake water replacement of 13% for each additional metre of lake depth ($R^2$ = 0.53, p < 0.001, Figure 6, Table 3). Lake water replacement was not independently correlated with any other lake or watershed

attribute including watershed area, lake volume, snow depth on the lake ice, lake ice thickness, lake area, and the ratio of lake area to watershed area (Table 3).

**Table 3.** Results for a linear regression between total lake water replacement with multiple lake and watershed properties. The adjusted $R^2$ and p-value are shown for each isotope. Linear regressions were performed using the 'lm' function using R 4.0.2 (R Core Team, 2021)

| | % Lake Volume Replaced by Freshet) | |
|---|---|---|
| Lake Attribute (unit) | Adjusted $R^2$ | p-value |
| lake depth (m) | 0.53 | <0.001 |
| lake area (m$^2$) | -0.06 | 0.849 |
| lake volume (m$^3$) | -0.03 | 0.486 |
| snow depth (cm) | -0.06 | 0.771 |
| ice thickness (m) | -0.05 | 0.654 |
| watershed area (m$^2$) | 0.02 | 0.274 |
| watershed area/lake area | -0.01 | 0.361 |

## 5   Discussion

### 5.1   Influence of snowmelt bypass on the replacement of lake water by freshet

Characterization of the influence of snowmelt bypass on the replacement of pre-snowmelt lake water required accurate deter-

mination of the isotopic composition of lake water prior to freeze-up. Given that lake water isotope samples are unavailable from Autumn 2017, and $\delta_{\text{L-Pre}}$ values were instead obtained from drilling through the lake ice before the lakes became ice-free, their isotopic compositions required correction for the isotope fractionation caused by ice formation. Our novel approach to correcting $\delta_{\text{L-Pre}}$ values for the fractionation caused by lake ice formation provides a reasonable estimate of $\delta_{\text{L}}$ prior to lake ice formation. While our correction of $\delta_{\text{L-Pre}}$ involves some uncertainty, such as having to estimate the relationship between lake

depth and lake volume, corrected $\delta_{\text{L-Pre}}$ values closely align with the general distribution of water isotope measurements from August and September 2018 of the same lakes (Figure C1). Corrected $\delta_{\text{L-Pre}}$ values are also situated near or above the LEL, reasonably indicating a more rainfall-sourced $\delta_{\text{I}}$ that would be present in lakes at the time of freeze-up during the previous

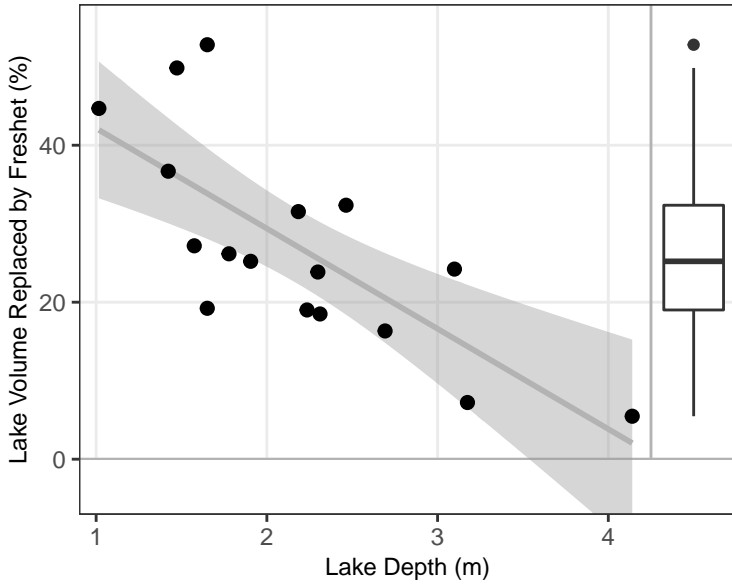

**Figure 6.** The relationship between the amount of lake water replaced by freshet and lake depth. The distribution of lake water replacement by freshet is shown by the boxplot on the right side of the plot. A linear regression is also displayed on the plot ($R^2 = 0.53$, $p < 0.001$). The shaded grey area represents the 95% confidence interval of the linear regression.

autumn. Prior to correction, most $\delta_{L\text{-Pre}}$ values plotted below the LEL (Figure 5a), indicating lakes had the majority of their inflow sourced from snow, which would be unlikely at the time when lake ice began forming during the previous autumn. We considered using $\delta_L$ values from September 2018 instead of correcting for ice fractionation, but 2018 was a cooler and wetter year than 2017, meaning the lake-specific $\delta_L$ values in September 2018 likely differed somewhat from September 2017.

The presence of a somewhat uniformly-thick layer of freshet beneath lake ice (Appendix E) likely explains the relationship between lake depth and the amount of lake water replaced by runoff, because the freshet layer represents a relatively smaller portion of lake volume at deeper lakes (Figure 6). We calculated the freshet layer thickness to be an average of 0.28 m, ranging from 0.12 to 0.52 m with a standard deviation of 0.11 m (Table E1). Previous studies have measured the thickness of the snowmelt bypass layer at the onset of freshet inflow to be ~30 – 200 cm (Henriksen and Wright, 1977; Bergmann and Welch, 1985). Since the mixed layer of pre-snowmelt lake water and freshet comprised a relatively larger volume in shallower lakes compared to deeper lakes (Figure 7a), a larger portion of lake water was able to be replaced with freshet in shallower lakes than in deeper lakes. We hypothesized that because shallower lakes likely had colder lakebed temperatures during freshet (Burn, 2005), more mixing between pre-snowmelt lake water and freshet inflow would occur due to the reduction in water density gradient between the bottom of the lake and the top of the lake. However, the estimated thickness of the freshet layer was uniform across lakes, indicating that colder lakebed temperatures may not have contributed to greater mixing at shallower lakes. To our knowledge, the relationship between lake depth and freshet retention has not been described in previous literature, although there has been little possibility to observe this relationship because estimates of freshet recharge in more than one lake

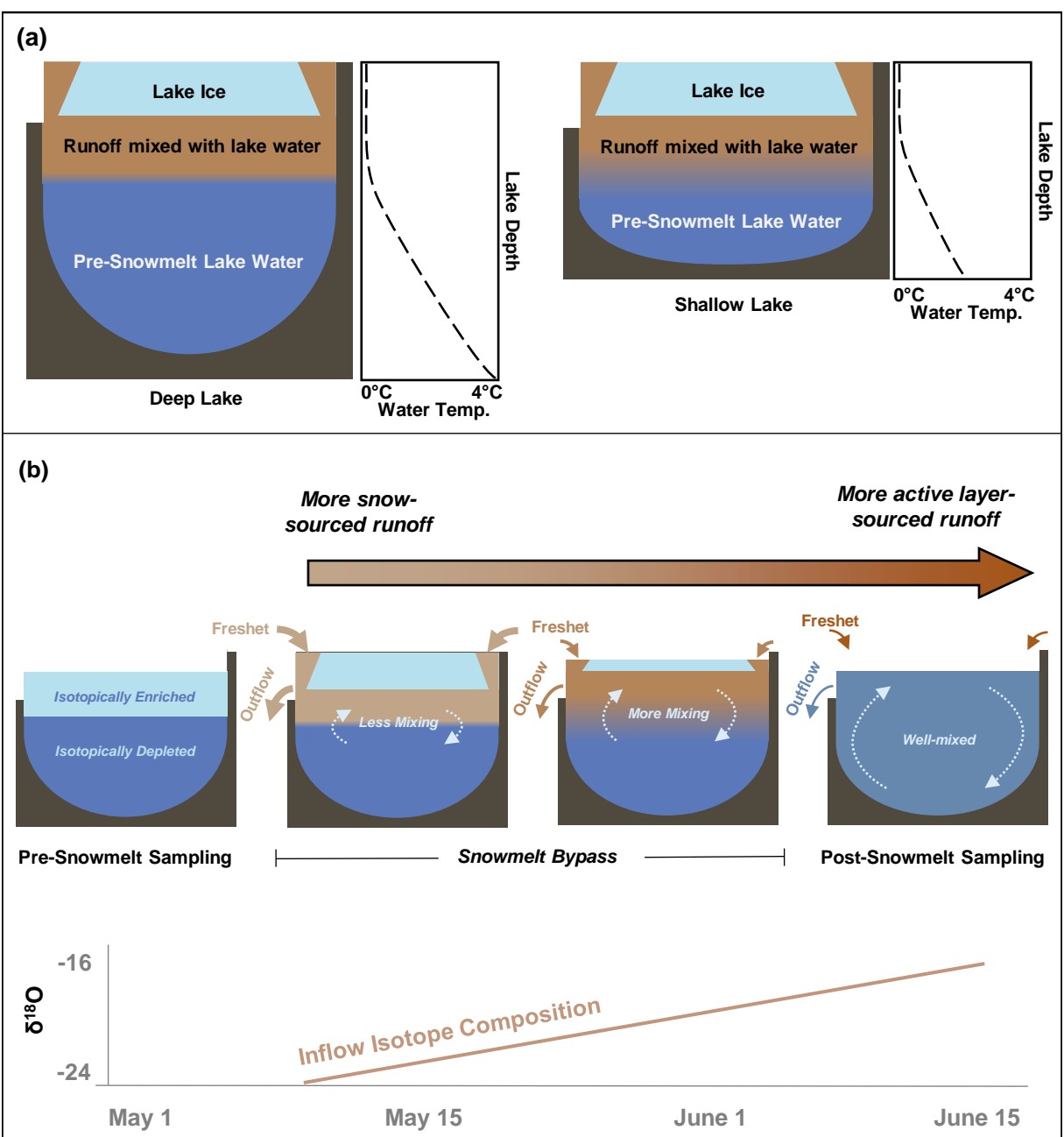

**Figure 7.** (a) A conceptual model showing the relative differences in snowmelt bypass between a shallow lake and a deep lake. Shallower lakes have a larger portion of their volume replaced by the runoff layer that flows beneath the ice, while a larger portion of water is isolated from mixing with runoff in deeper lakes. The estimated freshet layer thickness was somewhat uniform among all the lakes (Appendix E). (b) A conceptual model of how snowmelt bypass occurred over the course of the snowmelt period. Pre-snowmelt samples were taken from water beneath lake ice which was isotopically depleted in comparison to the lake ice. As time goes on, the source of freshet shifts from snow-sourced to active layer-sourced, while mixing increases beneath the lake ice simultaneously.

are scarce (Falcone, 2007; Brock et al., 2008). We expect that a similar relationship between lake depth and snowmelt bypass could be present in other open-drainage lakes that experience snowmelt bypass, since the relationship between increasing lake depth and greater snowmelt bypass is caused by the typical water temperature gradient which is present in ice-covered lakes at the onset of freshet. However, smaller lakes <1 ha, which are common in the Arctic (Pointner et al., 2019), likely do not experience as strong a snowmelt bypass effect because freshet is able to displace the pre-snowmelt lake water due to the small volume of the lake (Jansen et al., 2019; Cortés and MacIntyre, 2020). However, smaller lakes also typically have smaller under-ice chemical gradients that increase density stratification and limit freshet mixing and retention (Cortés and MacIntyre, 2020). Therefore, a relationship between lake depth and retention of freshet runoff likely does not exist at smaller arctic lakes.

## 5.2   Sources of freshet

Following the freshet, the $\delta_I$ of lakes did not shift towards the isotopic composition of snow ($\delta_{\text{Snow}}$) as one may expect, but instead shifted towards the average isotopic composition of precipitation ($\delta_P$, Figure 5c). Other than the 21.3 mm rainfall that fell during the six-week period between the two isotope sampling dates, the only other potential source of water during this time period is water stored in the active layer, which likely mixed with snowmelt runoff as the soil thawed throughout the spring. The high infiltration capacity of the peat channels that convey runoff causes nearly all snowmelt to flow through subsurface routes, as was observed in the field and reported by Quinton and Marsh (1998). As water near the surface of the active layer is most likely to be comprised of rainfall from the previous year, we expect that much of the water stored in the top of the active layer would have been largely sourced from rainfall from the previous summer and autumn. In support of this inference, Tetzlaff et al. (2018) reported $\delta^2H$ values between -140‰ and -160‰ from August to September of 2014 in water samples taken at 10 cm soil depth at Siksik Creek, a watershed directly adjacent to the Trail Valley Creek camp (Figure 2). This range of soil water $\delta^2H$ values is more enriched relative to $\delta_P$ ($\delta^2H$ = -160.1 ‰, Figure 5c), indicating that a mixture of snowmelt runoff with this active layer water could result in a water source similar in isotopic composition to $\delta_P$.

Freshet flowing into lakes later during the snowmelt period likely had a more rainfall-sourced isotopic composition, replacing the more snow-sourced runoff that had entered the lake earlier during the freshet (Figure 7b). A shift from snow-sourced water towards rainfall-sourced water during the course of the snowmelt period has been observed using water isotope measurements at Siksik Creek by Tetzlaff et al. (2018). Additionally, the mixing of freshet beneath lake ice increases with time as the temperature and density gradient lessens between the top and bottom of the lake water column (Cortés et al., 2017). Based on our results and these previous studies, we conclude that the ability of the active layer to contribute runoff to lakes appears to be maximized at the same time that vertical mixing in the lake is stronger, while snowmelt runoff flows into lakes at a time when little vertical mixing is occurring and is also likely replaced by later runoff (Figure 7b). Such interplay between timing of snowmelt runoff, lake ice melt and hydrological behaviour of the active layer explains why the source of water to lakes is not solely snow-sourced, and that incorporation of active layer runoff into lakes is more important than the volume of freshet delivered to lakes, for the open-drainage lakes of this study.

## 5.3 Assumptions and improving the utility of water isotope data from ice-covered lakes

In our estimation of lake water replacement by freshet we had to make some assumptions (Appendix C) when estimating $\delta_{\text{L-Pre}}$ using Equation 3. Future studies could sample lakes in the previous autumn before ice formation begins to avoid these assumptions, as there is minimal water flow in and out of arctic lakes during the winter months due to frozen soils and ice cover on lakes (Woo, 1980). If lakes cannot be sampled in the previous autumn, another option would be to take a lake ice core and sample the isotopic composition at different points along the lake ice core. These isotope measurements could then be used to estimate the $\alpha_{\text{eff}}$ value used in Equation 3, as has been done by Souchez et al. (1987) and Bowser and Gat (1995). This approach would avoid the assumptions we made in estimating $\alpha_{\text{eff}}$ outlined in Appendix C. However with this approach, one still needs to know the volume of the lake ice relative to the volume of the remaining unfrozen water, and must rely on lake bathymetry data or a depth–to–volume relationship, such as the one we derived using bathymetry data from Big Bear Lake (Equation 4). Since we only have one survey of lake bathymetry, we do not know how well our depth–to–volume relationship describes other lakes in the region, and it could be that this relationship varies as lakes increase in surface area or in areas of different surficial geology where the intensity of thermokarst processes may have varied during lake formation.

Since we do not have measurements of the lake temperature profile, we also assume our lakes have the typical thermal structure of cryomictic (Yang et al., 2021) ice-covered lakes that leads to snowmelt bypass. We relied on measurements of water temperature at 1.25 m depth in Big Bear Lake (Figure 2), and lake temperature profiles modelled using FLake-online (Appendix A) to conclude that our lakes were likely well-mixed at the time lakes were sampled post-freshet. Two scenarios were established in FLake-online, one representing a typical lake from the ones we sampled, and another lake representing a small, deep lake where mixing would be less likely. Even though FLake-online does not simulate under-ice mixing, which typically occurs in arctic lakes (Hille, 2015; Cortés et al., 2017; Kirillin et al., 2018), the 'typical lake' became fully mixed 1 day after becoming ice-free while the small, deep lake was simulated to become fully mixed 3 days after becoming ice-free. Even though snowmelt bypass is a common phenomenon in many types of ice–covered lakes around the world (Henriksen and Wright, 1977; Jeffries et al., 1979; Hendrey et al., 1980; Bergmann and Welch, 1985; Schiff and English, 1988; Edwards and McAndrews, 1989; Cortés et al., 2017), knowledge about how the thermal structure of our study lakes evolved over time, and varied between lakes of different depth, would have informed interpretation of our results. Such data could have helped better explain why shallow lakes retain more freshet runoff than deeper lakes, and also could have helped confirm our hypothesis that water flowing into lakes later during the freshet mixes more readily with lake water.

## 5.4 Climate change and snowmelt bypass

Future changes in snowmelt bypass are dependent on whether climate change allows open-drainage lakes to persist, or causes them to become closed-drainage, given that snowmelt bypass can only occur when lakes are at or above their outlet level. There are multiple consequences of Arctic warming that will influence lake water balance: changes in rainfall (Bintanja and Andry, 2017) and snowfall (Brown and Mote, 2009; Ernakovich et al., 2014), increases in active layer thickness (Walvoord and Kurylyk, 2016; Tananaev and Lotsari, 2022; Koch et al., 2022), the proliferation of deciduous shrubs (Loranty et al.,

2018), and longer lake ice-free periods (Woolway et al., 2020). Whether the combination of these changes will result in an increase or decrease in runoff to lakes is currently unknown (Blöschl et al., 2019), making it difficult to predict whether open-drainage lakes will persist or if some open-drainage lakes may shift to being closed-drainage under future climate. Due to this uncertainty, we discuss potential future changes in snowmelt bypass under two scenarios: a) where lakes remain open-drainage in the future, and b) where some lakes become closed-drainage in the future.

If open-drainage lakes persist as such in the future, we suspect the freshet that is incorporated into lakes may shift towards being more rainfall-sourced. Projected rainfall increases (Bintanja and Andry, 2017) will likely leave the active layer in a more saturated state when the active layer freezes in autumn, potentially providing more water to lakes during the freshet. Other studies have already established a strong positive relationship between increased rainfall in the previous summer and a more efficient conversion of the snowpack into freshet (Bowling et al., 2003; Stuefer et al., 2017), indicating the presence of a

strong connection between snowmelt runoff and water stored in the thawing active layer. Increasing shrub heights will advance snowmelt timing relative to lake ice breakup (Marsh et al., 2010; Wilcox et al., 2019; Grünberg et al., 2020), causing more snowmelt to flow into lakes at a time when there is limited below-ice vertical mixing. Combining earlier snowmelt timing with a more rain-saturated active layer could result in more freshet bypassing open-drainage lakes early during the freshet, with the active layer thawing deeper and shifting freshet more towards rainfall-sourced water by the time below-ice mixing begins.

Advancement of snowmelt timing and a shift from snow-sourced to rainfall-sourced freshet may have limnological implications for lakes. Cation and anion concentrations in snowmelt water tend to decrease over the course of the snowmelt period (Marsh and Pomeroy, 1999), while the pH of snowmelt runoff increases with time (Quinton and Pomeroy, 2006). Snowmelt also tends to have higher dissolved organic carbon (DOC) concentrations than summertime runoff (Finlay et al., 2006) and typically contributes the largest input of terrestrial organic mater to lakes in organic-rich landscapes (Townsend-Small et al.,

2011; Olefeldt and Roulet, 2012). Balasubramaniam et al. (2015) observed that thermokarst lakes dominated by snow-sourced water tended to have lower pH, higher conductivity and higher DOC concentrations than lakes dominated by rain-sourced water. Based on these observations, as snowmelt occurs earlier in the Arctic, lakes may experience decreases in DOC, and conductivity, and increases in pH. Such changes to lake biogeochemistry caused by shifts in freshet runoff retention could affect the productivity and ultimately the climate feedbacks of these lakes. Future research could combine estimates of lake water

replacement by freshet with water chemistry measurements to further our understanding of the impact of snowmelt bypass on lake chemistry and other limnological properties.

In a future where some open-drainage lakes become closed-drainage due to greater evaporation under a longer ice-free season, for example, we expect such lakes will retain more freshet runoff than comparable open-drainage lakes, because closed-drainage lakes retain the additional freshet that is required to recharge the lake to its outlet level. Since snowmelt bypass

cannot occur until a closed-drainage lake is recharged to its outlet level, we expect that freshet retention by closed-drainage lakes will not be as influenced by lake depth. Lakes with smaller ratios of watershed area to lake area (WA/LA) are more prone to a more negative water balance (Marsh and Pomeroy, 1996; Gibson and Edwards, 2002; Turner et al., 2014; Arp et al., 2015). Therefore, we expect lakes with relatively small WA/LA ratios will be more prone to becoming closed-drainage, relying on freshet to recharge them to their outlet level and retain more freshet as a result. A corollary of this prediction is that other

ice-covered lakes which currently lie below their outlet level at the onset of the freshet (i.e., closed-drainage lakes) likely retain more freshet than open-drainage lakes of a similar lake depth. A more saturated active layer at the onset of snowmelt, combined with a greater amount of snowfall should increase the ability of freshet to recharge any closed-drainage lake.

An additional complication to predicting the future of snowmelt bypass in response to climate change is caused by the shifting of lake ice regimes from bedfast ice to floating ice. Lakes that freeze completely to the bed in winter (bedfast ice) melt from the surface downwards in spring and likely do not develop the thermal stratification necessary for snowmelt bypass to occur. Remote sensing studies have already observed many bedfast ice lakes shifting to floating ice regimes during the past few decades in response to climate change (Arp et al., 2012; Surdu et al., 2014; Engram et al., 2018). We are unaware of any lake mixing studies on bedfast ice lakes, making it difficult to hypothesize about how shifting from bedfast to floating ice could affect the amount of freshet retained by a lake.

## 6  Conclusions

A portion of the large volume of freshet that flows into lakes every year can bypass ice-covered, open-drainage lakes due to limited mixing between lake water beneath the lake ice and freshet. By estimating the percentage of lake water replaced by freshet at 17 open-drainage lakes, we have been able to explore which lake and watershed attributes affect snowmelt bypass. Our data show that as lake depth increases the amount of lake water replaced by freshet decreases, likely because freshet is unable to mix with deeper lake water when lakes are ice-covered and the water column is stratified, however we lack data demonstrating the extent of mixing in the lakes we studied. Additionally, the volume of freshet flowing into the lakes seems to have minimal impact on the amount of lake water replaced by freshet, given that the ratio of watershed area to lake area was not correlated with the percentage of lake water replaced by freshet. Estimation of the isotopic composition of source waters showed that the freshet remaining in lakes was not solely snow sourced – rainwater left in the active layer from the previous autumn had mixed with snowmelt before entering lakes. Active layer-sourced water likely flows into lakes later in spring and at a time when freshet can more easily mix with pre-snowmelt lake water, replacing the earlier more snow-sourced freshet.

Models specialized for northern environments are rapidly improving their ability to represent the complicated processes present in permafrost regions, such as the effect of shrubs on snow accumulation, snowmelt and active layer thickness (Krogh and Pomeroy, 2019; Bui et al., 2020), lake ice formation and decay (MacKay et al., 2017) and the mixing processes that lead to snowmelt bypass (MacKay et al., 2017). Such models could be used to examine how freshet water sources may change in the future, which could have significant impacts on limnological properties including water chemistry (Finlay et al., 2006; Balasubramaniam et al., 2015). Additionally, current physically-based lake models can represent vertical mixing beneath lake ice (MacKay et al., 2017), and could be used to further evaluate the influence of lake depth, lake ice regime, or climate change on snowmelt bypass and resulting impacts on limnology.

*Data availability.* The data used in the paper are presented in tables in the manuscript and Appendix B and D. Isotope data and lake and watershed attribute data can be downloaded from the Trail Valley Creek Research Station Dataverse at https://doi.org/10.5683/SP3/AZE4ER. Meteorological data were retrieved from Environment and Climate Change Canada at https://climate.weather.gc.ca/historical_data/search_historic_data_e.html.

## Appendix A: Lake mixing status at ice-off: water temperature data and modelling

The application of Equation 2 assumes that lakes are well-mixed at the time that $\delta_{\text{L-Post}}$ water samples were corrected. We investigated if the lakes we sampled were likely to have been well-mixed at ice-off, because some lakes have been observed to be not well-mixed at ice-off (Vachon et al., 2019; Wiltse et al., 2020; Cortés and MacIntyre, 2020). We rely on water temperature data from Big Bear Lake and another lake near the Inuvik-Tuktoyaktuk Highway investigated by Hille (2015), and lake temperature modelling using FLake-online (Kirillin et al., 2011).

Water temperature measurements at Big Bear Lake and a lake nearby the Inuvik-Tuktoyaktuk Highway suggest that lakes in this region become well-mixed during the ice-off period. At Big Bear Lake, water temperature at 1.25 m depth reaches 4°C initially by 05-06-2018, followed by daily fluctuations between 2.3 – 4.1°C, before continuing a warming trend again on 13-06-2018 and reaching 6.8°C on 15-06-2018 (Figure A1). Water temperature between 0 and 4 m was measured by Hille (2015) at a 10 m deep lake approximately 10 km from the Inuvik-Tuktoyaktuk highway in 2009. Hille (2015)'s measurements

show uniform warming of the water column beneath the ice from 1 to 4 m, with water temperatures reaching ~5°C by the time the lake became ice-free (Hille, 2015, Figure 3.4). Based on these observations, we assume these two lakes were well-mixed at the time they became ice-free.

We also ran two model scenarios using FLake-online (Kirillin et al., 2011, http://flake.igb-berlin.de/model/run) to gather further information about the mixing status of lakes in this region at the time they become ice-free. The first scenario represents

a typical lake compared to the lakes we sampled in size and depth (Table A1). In the second scenario, lake depth is increased and lake area is decreased to represent the "worst-case scenario" for lake mixing after lakes became ice-free (Table A1). Water clarity was set to 2 m, based on an average Secchi depth measurement of 1.88 m based on measurements made by Vucic et al. (2020) at lakes along the Inuvik-Tuktoyaktuk Highway and Dempster Highway south of Inuvik. Both scenarios were run with 'perpetual year' meteorological forcing data, whereby meteorological data from 01-11-2015 to 31-10-2016 forces the model

for multiple years until a quasi-steady equilibrium is reached.

In both model scenarios, the mixing depth reached the maximum lake depth rapidly after ice-off, taking one day in the typical lake scenario and three days in the worst chance of mixing scenario. Notably, under-ice warming and mixing does not occur in FLake-online, although under-ice mixing has been observed in other Arctic lakes (Hille, 2015; Cortés et al., 2017; Kirillin et al., 2018). Given that both models indicate full mixing within a few days of lakes becoming ice-free, despite the absence of

405 under-ice warming and mixing that was likely present at our study lakes based on temperature data from Big Bear Lake and Hille (2015), we believe that the lakes we sampled were very likely fully-mixed when we sampled them on 2018-06-15.

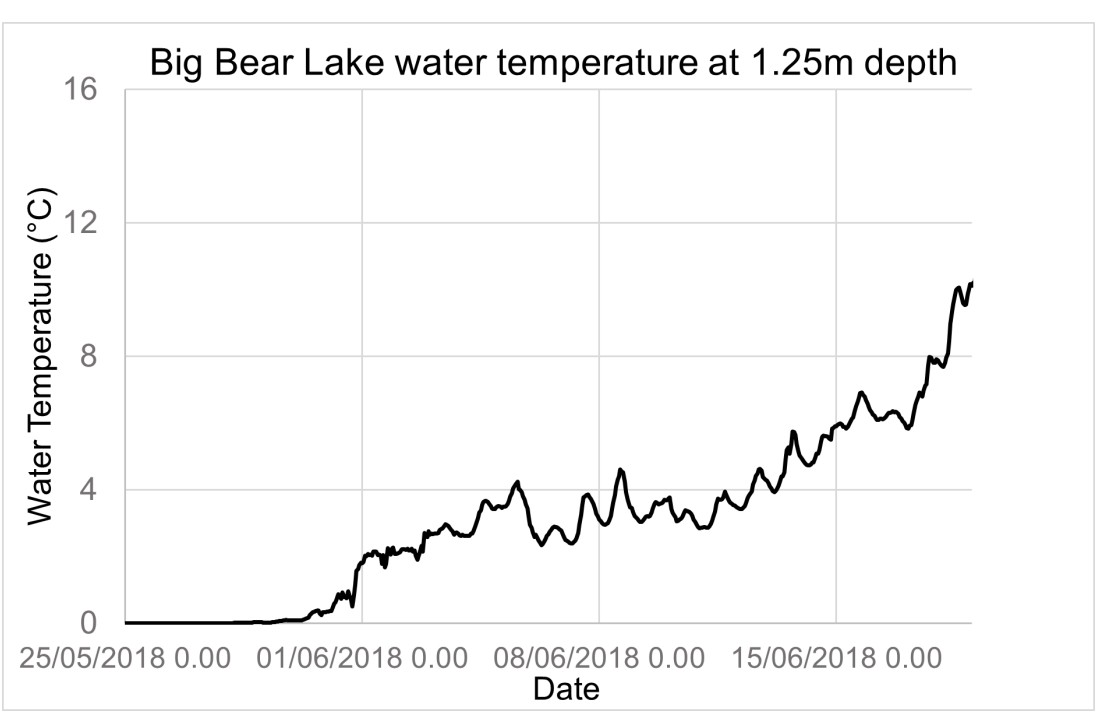

**Figure A1.** Big Bear Lake water temperature at a depth of 1.25 m, between 2018-05-25 and 2018-06-19.

**Table A1.** Input parameters and results for the two scenarios used to evaluate the mixing conditions of lakes during the ice-off period.

| Parameter/Result | Typical Lake Scenario | Worst Chance of Mixing Scenario |
|---|---|---|
| Coordinates | 68.74, -133.53 | 68.74, -133.53 |
| Mean Lake Depth (m) | 2 | 3.5 |
| Water transparency (m) | 2 | 2 |
| Lake Fetch (m) | 250 | 50 |
| Ice-free Date | June 28 | June 28 |
| Fully-mixed Date | June 29 | July 1 |
| Days from Ice-free to Fully-mixed | 1 | 3 |

## Appendix B: Isotope framework and sensitivity analysis

**Table B1.** Variables used in isotope framework and sources of their calculation.

| Parameter | Value | Reference |
|---|---|---|
| $\delta^*$ (‰) | $\delta^{18}O = -10.77$, $\delta^2H = -104.97$ | (Gonfiantini, 1986) |
| $h$ (%) | 80.5 | (Environment and Climate Change Canada, 2019) |
| $T$ (K) | 282.32 | (Environment and Climate Change Canada, 2019) |
| $\alpha^*_{L-V}$ | $^{18}O = 1.0108$, $^2H = 1.0981$ | (Horita and Wesolowski, 1994) |
| $\varepsilon^*$ | $^{18}O = 0.0108$, $^2H = 0.0981$ | (Horita and Wesolowski, 1994) |
| $\varepsilon_k$ | $^{18}O = 0.0028$, $^2H = 0.0024$ | (Gonfiantini, 1986) |
| $\delta_{Rain}$ (‰) | $\delta^{18}O = -16.79$, $\delta^2H = -128.7$ | This study. |
| $\delta_{Snow}$ (‰) | $\delta^{18}O = -24.61$, $\delta^2H = -184.2$ | This study. |
| LMWL Slope, Intercept (‰) | 7.066, $\delta^2H = -10.0$ | This study. |
| LEL Slope , Intercept | 5.114, 48.9 | This study. |

The isotope framework (i.e., establishment of the predicted Local Evaporation Line (LEL)) used for this study was based on the coupled isotope tracer method developed by Yi et al. (2008), following other studies that have investigated lake water balances using water isotope tracers (Turner et al., 2014; Remmer et al., 2020; MacDonald et al., 2021). Below are the parameters and equations required to calculate $\delta^*$, the terminal point on the LEL. The equation for $\delta^*$, which represents the isotopic composition of a lake at the point of desiccation, is as follows (Gonfiantini, 1986):

$$\delta^* = \frac{h * \delta_{As} + \varepsilon_k + (\varepsilon^*/\alpha^*)}{h - \varepsilon_k - (\varepsilon^*/\alpha^*)} \tag{B1}$$

where $\alpha^*$ is the fractionation factor between the liquid and vapour phase of water (Horita and Wesolowski, 1994), calculated for $\delta^{18}O$ as:

$$\alpha^*_{L-V} = 2.718^{(-7.685 + 6.7123 * \frac{10^3}{T} - 1.6664 * \frac{10^6}{T^2} + 0.35041 * \frac{10^9}{T^3})/1000} \tag{B2}$$

and calculated for $\delta^2H$ as:

$$\alpha^*_{L-V} = 2.718^{(1158.8 * \frac{T^3}{10^9} - 1620.1 * \frac{T^2}{10^6} + 794.84 * \frac{T}{10^3} - 161.04 + 2.9992 * \frac{10^9}{T^3})/1000} \tag{B3}$$

The term $\varepsilon^*$ is a separation term where:

$$\varepsilon^* = \alpha^* - 1 \tag{B4}$$

The term $h$ represents the relative humidity of the air above the water and $\delta_{As}$ is the isotopic composition of atmospheric moisture during the open water season defined as:

$$\delta_{As} = \frac{\delta_{Ps} - \varepsilon^*}{\alpha^*} \tag{B5}$$

where $\delta_{Ps}$ is the average isotopic composition of precipitation (i.e., rainfall) during the open water season. The term $\varepsilon_k$ is the kinetic fractionation separation term, defined as

$$\varepsilon_k = x * (1 - h) \tag{B6}$$

where x = 0.0142 for $\delta^{18}O$ and x = 0.0125 for $\delta^2H$ (Gonfiantini, 1986).

Given that there is some variability in $\delta_{Snow}$ and $\delta_{Rain}$ values from samples collected, we conducted a sensitivity analysis to evaluate whether uncertainty in these values could affect $\delta_I$ and % lake water replacement sufficiently to change our interpretation of the results. To conduct the sensitivity analysis, we calculated the standard error of the mean (SEM) for $\delta_{Snow}$ and $\delta_{Rain}$:

$$SEM = \frac{\sigma}{\sqrt{n}} \tag{B7}$$

where $\sigma$ is the standard deviation of $\delta_{Snow}$ or $\delta_{Rain}$ values and $n$ is the number of $\delta_{Snow}$ or $\delta_{Rain}$ samples. The SEM was added to $\delta_{Snow}$ and $\delta_{Rain}$ to calculate an "upper bound" estimate, and subtracted to calculate a "lower bound" estimate. These upper and lower bound $\delta_{Snow}$ and $\delta_{Rain}$ values were then used to calculate upper and lower bound $\delta_P$, $\delta_{Ps}$ and $\delta^*$ values (Table B2). These upper and lower bound values were then used to calculate upper and lower bound $\delta_I$ (Figure B1a) and % lake water replaced values (Figure B1b) were calculated. Overall, $\delta_I$ and % lake water replaced values change minimally between the upper and lower bound cases (Figure B1a, B1b), and do not alter our interpretation of the results.

**Table B2.** Comparison of isotope framework parameters for upper and lower bound cases.

| Case | $\delta^{18}O_P$ | $\delta^2H_P$ | $\delta^{18}O_{Ps}$ | $\delta^2H_{Ps}$ | $\delta^{18}O^*$ | $\delta^2H^*$ |
|---|---|---|---|---|---|---|
| Lower Bound | -24.55 | -181.12 | -17.40 | -133.46 | -11.36 | -109.40 |
| Base | -24.10 | -178.00 | -16.79 | -129.15 | -10.78 | -104.97 |
| Upper Bound | -23.65 | -174.88 | -16.18 | -124.84 | -10.16 | -100.54 |
| SEM | ±0.45 | ±3.12 | ±0.61 | ±4.31 | ±0.61 | ±4.43 |

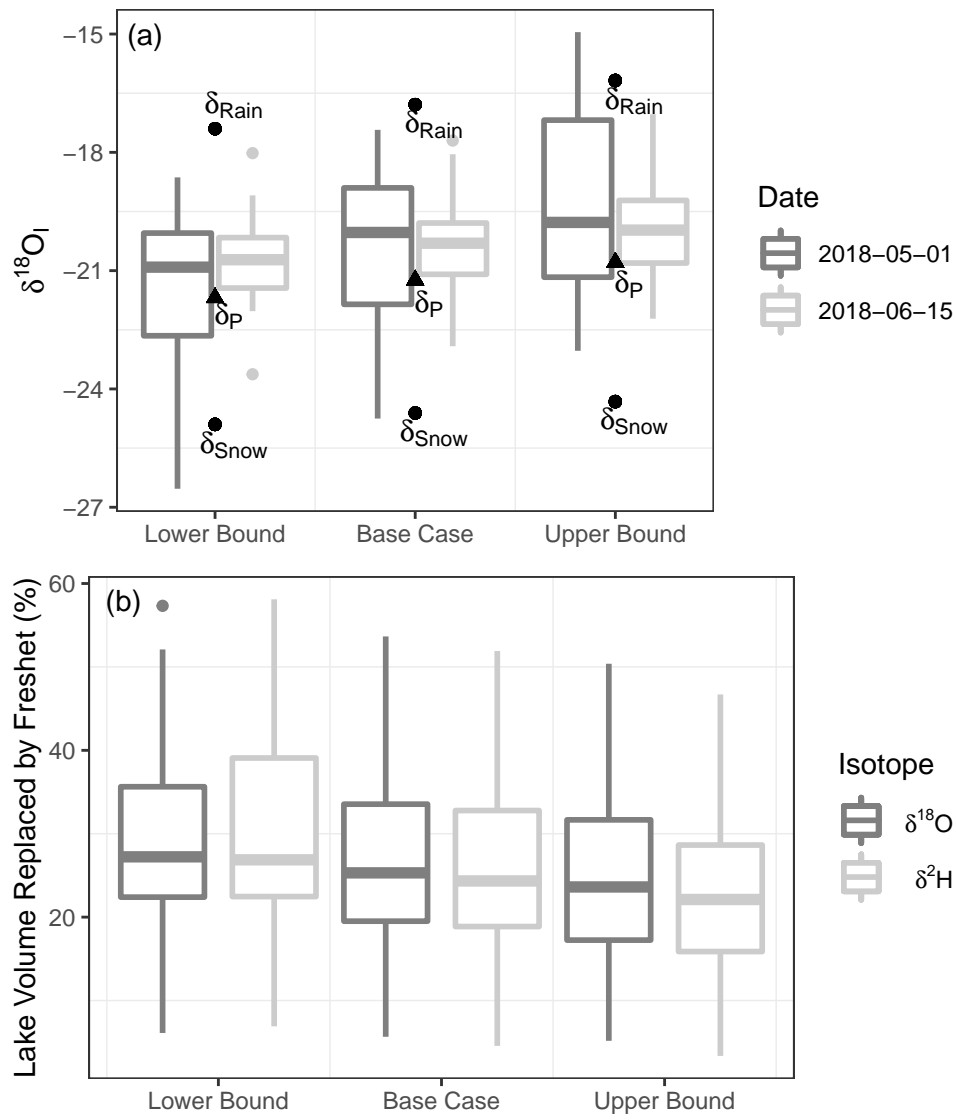

**Figure B1.** Comparison of upper and lower bound $\delta_I$ and % lake water replacement values against the base case.

## Appendix C: Determination of $\alpha_{\text{eff}}$ values

In order to determine $\alpha_{\text{eff}}$, two variables must be taken into account: the thickness of the $^{18}$O or $^2$H boundary layer across which heavy isotopes are diffusing from water into ice, and the downward velocity of the freezing ice (Ferrick et al., 2002). If these two parameters are known, the fractionation factor can be estimated using a linear resistance model developed by Ferrick et al. (1998), which is similar in structure to the Craig and Gordon (1965) linear resistance model for evaporation. Ferrick et al. (1998) define the effective fractionation factor between ice and water as:

$$\alpha_{eff} = \frac{\alpha^*_{L-S}}{\alpha^*_{L-S} + (1 - \alpha^*_{L-S})exp[\frac{-zv}{D_i}]} \tag{C1}$$

where $\alpha^*_{L-S}$ is the equilibrium fractionation factor between ice and water (1.002909 for $^{18}$O/$^{16}$O, 1.02093 for $^2$H/$^1$H (Wang and Meijer, 2018)), $z$ is the $^{18}$O or $^2$H boundary layer thickness between the ice and water (mm), $v$ is the velocity of ice growth (cm$^2$ day$^{-1}$), and $D_i$ is the self-diffusion coefficient of $^1$H$_2$$^{18}$O or $^1$H$^2$H$^{16}$O at 0°C (cm$^2$ day$^{-1}$). As the boundary layer and the velocity of ice growth increase, $\alpha_{\text{eff}}$ moves from the value of $\alpha^*_{L-S}$ towards a value of 1 (i.e., no fractionation).

As we do not know the boundary layer thickness at the ice-water interface, or the exact ice growth velocity for the lakes studied here, we relied on multiple other sources of information to estimate a probable upper and lower bound of $\alpha_{\text{eff}}$. We took into account previous estimates of $\alpha_{\text{eff}}$ for ice-water fractionation (Souchez et al., 1987; Bowser and Gat, 1995; Ferrick et al., 2002) and boundary layer thickness from other studies of lakes (Ferrick et al., 1998, 2002; Gibson and Prowse, 2002). The boundary layer thickness between water and freezing ice in a lake was estimated to be between 1 mm and 6 mm by Ferrick

et al. (1998), however, they revised this estimate with a more rigorous diffusion model to 1 ±0.3 mm for $^{18}$O and 0.4 ± 0.2 mm for $^2$H (Ferrick et al., 2002). They also found that the boundary layer thickness remained mostly stable across different ice growth velocities, although the lowest ice growth velocity of ~0.9 cm day$^{-1}$ had a boundary layer of ~1.8 mm (Ferrick et al., 2002). The mean $^{18}$O $\alpha_{\text{eff}}$ values for two ice cores taken from the lake studied by Ferrick et al. (2002) were 1.0021 and 1.0020, with respective ice growth velocities of 3.7 and 4.1 cm day$^{-1}$. A 1 mm boundary layer was also estimated by Gibson and Prowse

(2002) beneath river ice in northern Canada, however they also suggest that the boundary layer thickness can reach up to 4 mm in quiescent lake water. Therefore, we assume a minimum boundary layer thickness of 1 mm, and a maximum boundary layer thickness of 4 mm.

We estimated the minimum possible freezing velocity of our study lakes using the initial date of ice formation and the ice thickness we measured in spring. Based on Sentinel imagery (Sentinel Playground), all studied lakes became ice-covered by

October 16th, 2017. Ice thickness was measured at Big Bear and Little Bear Lake (near Trail Valley Creek camp, Figure 1) on March 21st, 2018, and when ice thickness was remeasured again in late April, it had not become any thicker. Therefore, assuming ice growth began on October 16, 2017 and ceased on March 21, 2018, the ice growth velocities for our study lakes range from an average of 0.50 – 0.84 cm day$^{-1}$ (0.78 – 1.32 m ice thickness). This only provides a lower bound estimate for ice growth velocity, as ice growth likely stopped earlier than March 21, 2018, and was more rapid during initial ice formation.

We further constrained our estimate of $\alpha_{\text{eff}}$ by assuming that $\alpha_{\text{eff}}$ values that result in lake water replacement estimates of >100% or <0% were not correct. Using all these sources of information, we calculated an upper bound of $\alpha_{\text{eff}}$ values based on

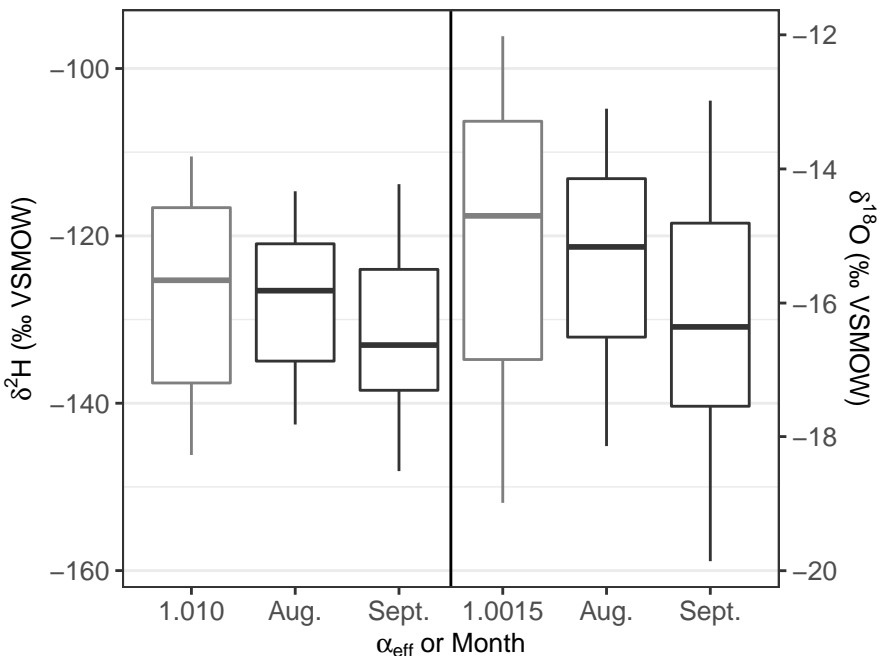

**Figure C1.** Comparison of Pre-Lake Ice Formation and August / September 2018 isotopic composition of lake waters. The $\alpha_{\text{eff}}$ values ($\delta^2$H $\alpha_{\text{eff}}$ = 1.010, $\delta^{18}$O $\alpha_{\text{eff}}$ = 1.0015) match closely with the August and September 2018 isotopic composition of lake waters. This suggests that these $\alpha_{\text{eff}}$ estimates are appropriate to use for estimating the pre-ice formation isotopic composition of lake waters.

the minimum possible ice freezing velocity ($^2$H $\alpha_{\text{eff}}$ = 1.0199, $^{18}$O $\alpha_{\text{eff}}$ = 1.00286) and lower bound of $\alpha_{\text{eff}}$ values which still generate lake water replacement estimates that are >0% and <100% ($^2$H $\alpha_{\text{eff}}$ = 1.010, $^{18}$O $\alpha_{\text{eff}}$ = 1.0015).

Assuming a 2 mm boundary layer, which is within the range of our boundary layer thickness estimates, $\alpha_{\text{eff}}$ values of 1.0015 for $^{18}$O and 1.010 for $^2$H correspond to ice growth rates of 3.62 cm day$^{-1}$ for $^{18}$O and 3.34 cm day$^{-1}$ for $^2$H. Similar ice growth rates have been observed in Arctic lakes (Woo, 1980); greater ice growth rates were also estimated for a lake in a warmer climate by Ferrick et al. (2002). These $\alpha_{\text{eff}}$ values also compare well with other measured $\alpha_{\text{eff}}$ values in lakes: a range of $\alpha_{\text{eff}}$ = 1.013 – 1.015 for $^2$H was found by Souchez et al. (1987) for a 4.4 cm thick lake ice cover; $\alpha_{\text{eff}}$ has been found to range from 1.0005 to 1.0027 for $^{18}$O in a single 50 cm ice core (Bowser and Gat, 1995). The $\delta_{\text{L-Pre}}$ values calculated using $\alpha_{\text{eff}}$ =

1.0015 for $^{18}$O and 1.010 for $^2$H also closely match the distribution of $\delta_{\text{L}}$ values from August and September of 2018, giving an indication that these $\alpha_{\text{eff}}$ values generate realistic isotopic compositions of lake water (Figure C1). Therefore, we chose $\alpha_{\text{eff}}$ = 1.0015 for $^{18}$O and $\alpha_{\text{eff}}$ = 1.010 for $^2$H as our $\alpha_{\text{eff}}$ values, as they correspond well with other estimates $\alpha_{\text{eff}}$, are within a range of probable ice-growth rates and lake water replacement by freshet and generate pre-ice formation isotopic compositions that closely match the following summer's isotopic composition of lake waters.

 **Appendix D: Bathymetric data and volume – depth relationship**

**Table D1.** Big Bear Lake bathymetric data and fit between modelled relationship between lake volume and lake depth as a percentage of total lake volume and depth.

| Depth (m) | Cumulative Depth (% total) | Cumulative Volume (m$^3$) | Cumulative Volume (% total) | Modelled Cumulative Volume (% total) | Offset between data and modelled cumulative volume (%) |
|---|---|---|---|---|---|
| 2.5 | 100 | 89326.83 | 100 | 100 | 0.00 |
| 2.25 | 90 | 88327.68 | 98.88 | 99 | -0.12 |
| 2 | 80 | 85881.37 | 96.14 | 96 | 0.14 |
| 1.75 | 70 | 82441.90 | 92.29 | 91 | 1.29 |
| 1.5 | 60 | 77491.01 | 86.75 | 84 | 2.75 |
| 1.25 | 50 | 69384.35 | 77.67 | 75 | 2.67 |
| 1 | 40 | 58991.30 | 66.04 | 64 | 2.04 |
| 0.75 | 30 | 46548.63 | 52.11 | 51 | 1.11 |
| 0.5 | 20 | 32463.34 | 36.34 | 36 | 0.34 |
| 0.25 | 10 | 16944.40 | 18.97 | 19 | -0.03 |
| 0 | 0 | 0 | 0 | 0 | 0.00 |

The modelled relationship between lake depth and volume is:

$$\%_{LakeVolume} = (-0.01 * \%_{LakeDepth})^2 + 2 * \%_{LakeDepth} \tag{D1}$$

where $\%_{\text{LakeVolume}}$ is the cumulative lake volume as a percent of total and $\%_{\text{LakeDepth}}$ is the cumulative lake depth as a percent of total (Table D1).

 **Appendix E: Freshet layer thickness**

We computed the thickness of the freshet layer using the relationship between lake depth and lake volume, the % of lake volume replaced by freshet, and lake depth measurements made at each sample lake. The freshet layer thickness was calculated by rearranging Equation 5:

$$freshet\,layer\,thickness\,(m) = \frac{lake\,depth}{-10 * \sqrt{100 - \%lake\,water\,replaced} - 10} \tag{E1}$$

This calculation represents the thickness the freshet layer would be on 2018-06-15 if it had not mixed with pre-snowmelt lake water. In reality, the unmixed freshet layer during the height of freshet would likely be thicker, due to rises in lake level caused by freshet that are not accounted for in this equation. Layer thicknesses averaged 0.28 m, ranging from 0.12 m to 0.52 m with a standard deviation of 0.11 m (Table E1).

**Table E1.** Calculated layer thickness for each lake using Equation E1

| Lake | Freshet Layer Thickness (m) |
|------|------|
| 7 | 0.23 |
| 8 | 0.29 |
| 9 | 0.26 |
| 10 | 0.17 |
| 11 | 0.26 |
| 14 | 0.29 |
| 15 | 0.23 |
| 16 | 0.12 |
| 19 | 0.44 |
| 20 | 0.23 |
| 21 | 0.25 |
| 26 | 0.43 |
| 27 | 0.40 |
| 49 | 0.38 |
| 50 | 0.52 |
| 51 | 0.22 |
| 52 | 0.12 |
| Min | 0.12 |
| Mean | 0.28 |
| Max | 0.52 |
| St. Dev. | 0.11 |

*Author contributions.* E.J.W developed the study design and sampling plan with input from B.B.W and P.M. E.J.W completed field sampling and sample preparation for lab analysis. E.J.W completed the data analysis with input from B.B.W. E.J.W lead the writing of the manuscript with input from B.B.W and P.M.

*Competing interests.* The authors declare that they do not have any conflict of interest.

*Acknowledgements.* E.J.W was funded by a W. Garfield Weston Award for Northern Research and Ontario Graduate Scholarship. We acknowledge funding from ArcticNet, Northern Water Futures, Northern Scientific Training Program, Polar Continental Shelf Program, the Canada Research Chairs program, and the Natural Sciences and Engineering Research Council of Canada. The research license (No. 16237)

was administered by the Aurora Research Institute in Inuvik, Northwest Territories, Canada, and can be found at http://data.nwtresearch. com/Scientific/16237. The authors are grateful for the assistance of the Arctic Hydrology Research Group members for their help in completing the field work. The authors also thank the staff of the UW-EIL for completing the isotope analyses. We also would like to thank the anonymous reviewers who helped improve the quality and clarity of this manuscript.

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
