# Peer review of "Assessing the influence of lake and watershed attributes on snowmelt bypass at thermokarst lakes"

_Hydrology and Earth System Sciences, 2022_

## Author Response (AR1)

**HESS-2022-133 Response to Reviewers**

September 16, 2022

Reviewer 1
*This study by Wilcox et al. uses established stable isotope tracers of lake hydrological processes in an innovative way in order to assess snowmelt bypass and retention in multiple representative open-drainage thermokarst lakes. A key result of the study is that the spring freshet replaces a larger fraction of lake water in shallower lakes, while this replacement does not appear to depend on catchment characteristics. The isotope tracers also allowed the authors to identify contributions of different meltwater sources (snowpack or rain / thawing active layer), which is an important result given the impact of meltwater provenance on lake biogeochemistry.*

Thank you for your review of our study, we appreciate your thoroughness which has made the results of this study more robust and transparent.

*General comments*

*In the introduction, the reader would be aided by a broader discussion of the rationale of the study. Why would one expect the snowmelt bypass to be a function of lake and watershed characteristics? Which characteristics are important and why? And what will this information enable us to do next?*

- We agree that the rationale of this study was a bit opaque. We have added detail about why one would expect lake characteristics, such as depth and watershed size would affect snowmelt bypass given the effect of lake depth on thermal profiles of lakes and the ability of incoming freshet to displace pre-snowmelt lake water (as you highlighted in another comment) in the Introduction. We also added that knowledge of how lake and watershed attributes affect snowmelt bypass allows us to better interpret and understand why water chemistry and limnological properties may vary from lake to lake in a given region.

*It may be worthwhile to briefly discuss evidence that smaller lakes and basins (≤ 1 ha) can be flushed entirely with meltwater in a matter of days (Jansen et al., 2019; Cortés and MacIntyre, 2020). This process likely affects a majority of Arctic lakes and ponds, which tend to be very small, and a brief discussion of the literature would extend the findings of this paper.*

- Thank you for pointing this out as we had skimmed over some of these important aspects of lake mixing in our Introduction. We have now added this information to the Discussion.

*Uncertainty about the isotopic end members: Figure 5c indicates that considerable uncertainty exists about the mean isotopic composition of rain and the snow pack and thus $\delta_P$. This implies that omitting one or several of the rain or snow samples from the analysis – or including additional samples - could significantly change the results and their interpretation. How would the uncertainty about $\delta_{rain}$, $\delta_{snow}$ and $\delta_P$ affect the uncertainty about the estimates of $\delta^*$, $\delta_I$ and % lake water replaced? I strongly recommend that the authors compute the error propagation for the quantities affected, or conduct a sensitivity analysis.*

- We somewhat disagree that there is "considerable uncertainty" about the mean isotope composition of the mean snowfall and rainfall isotope composition. The isotope composition of any given precipitation (rain or snow) event will naturally vary from the average, and the degree of variability observed in our rain and snow samples is comparable to other studies carried out in the Arctic. While the wide range of rainfall and snowfall isotope compositions may indicate uncertainty when one has a small sample size relative to the "population size" (number of rainfall events), when the population size is close to the sample size, the uncertainty is lower. Given that we were able to sample the most of the rainfall events during the sampling period, we believe that $\delta_{Rain}$ is well estimated. Additionally, the close plotting of lake water isotope compositions to the local evaporation line (LEL) between $\delta_P$ and $\delta_{SSL}$ provides some suggestion that $\delta_P$ is well estimated, although not all lakes will plot along the LEL if they are heavily influenced by snow-sourced or rain-sourced waters.

- We agree though that incorporating sensitivity analysis into our methods does strengthen one's ability to trust the estimated values of $\delta_I$ and %lake water replaced, and have incorporated such an analysis into Appendix A, where the isotope framework is described. We now apply a straight-forward error propagation analysis by first calculating the standard error of the rainfall and snowfall samples (standard error of mean = standard deviation/ sqrt(n)), and then we calculate the upper bound and lower bound $\delta_P$ by adding or subtracting the standard error from the mean. The standard error of $\delta_P$ is $\delta^{18}O$ = -21.24± 0.45 and $\delta^2H$ = -160.103± 3.118, while the standard error of $\delta^*$ is $\delta^{18}O$ = -10.78±0.62‰ and $\delta^2H$ = -104.97±4.310‰. These upper and lower bounds $\delta^*$ are then used to calculate upper and lower bound $\delta_I$ and %lake water replaced. The upper and lower bound $\delta_I$ values are compared against the respective upper and lower bound $\delta_P$, $\delta_{Rain}$, and $\delta_{Snow}$ values to evaluate if our interpretation of the changes in $\delta_I$ values before and after snowmelt would differ if we used the upper or lower bound cases. Ultimately, whether one uses the upper or lower bounds does not change the interpretation of the data provided in the manuscript, which is why we include the sensitivity analysis results in the Appendix.

*Assumption of mixing: the computation of the percentage of lake water replaced by freshet requires that the lakes be fully mixed during sampling immediately after ice-off. However, some (shallow) arctic and boreal lakes have been shown to mix incompletely in spring, especially when thermal stratification is strengthened by chemical stratification, or prior to wind mixing just after ice-off (Vachon et al., 2019; Wiltse et al., 2020; Cortés and MacIntyre, 2020). I think it is important that the authors provide evidence that the lakes were fully mixed during sampling on June 15 2018, for example by showing temperature profile observations at the time of sampling in a representative lake along the transect, or by applying a simple 1D hydrodynamic model such as FLake (http://www.flake.igb-berlin.de/).*

- The assumption that lakes are well-mixed at ice-off is definitely an assumption that we had overlooked in this study.

- Apologies to the editor who asked if we had temperature profile data for the same reason, and at the time we did not think we had any temperature data from lakes. We now remembered that a water level recorder in Big Bear Lake (max depth 2.5 m) also recorded temperature, at a water depth of ~1.25m. Water temperature at this depth reaches 4°C initially by 05-06-2018, followed by daily fluctuations between 2.3 – 4.1°C, before continuing a warming trend again on 13-06-2018 and reaching 6.8°C on 15-06-2018.

- Additionally, we found a lake water temperature profile measured by Hille (2009) at a 10.9 m deep lake 10 km of the Inuvik-Tuktoyaktuk highway. Hille's data (Figure 3.4) show uniform warming of the water temperature between 1 and 4 metres from ~2°C to ~5°C by the time the lake became ice-free.

- We also ran "FLake-Global: online modelling system" (http://flake.igb-berlin.de/model/run) using two different sets of parameters representing a "typical" lake from our dataset, and for a small/deep lake where we would expect the least favourable conditions for mixing after lakes became ice-free. We opted for a "perpetual year" run where meteorological data from 01-10-2015 to 30-09-2016 is used to force the model for multiple years until a quasi-stationary state is reached. The model parameters for the 'typical' lake is based roughly on averages taken from Table 1 in the manuscript: lake fetch was set at 250 m (a perfectly circular lake of this fetch would be 5 ha, close to the median lake size of 4.8 ha), mean lake depth was set to 2 m, and the coordinates used were the average coordinates of the lakes we sampled. The water type was "clear (2m transparency) based on an average Secchi depth of 1.88 m measured at lakes along the Inuvik-Tuktoyaktuk Highway and Dempster highway south of Inuvik by Vucic et al. (2020). This model run simulates the mixed layer depth reaching the maximum lake depth within one day of the lake becoming ice-free. We ran the second "worst chance of mixing" scenario with a mean depth of 3.5 m, and a fetch of 50 m, with other parameters identical to the first scenario. In this scenario, the mixed layer reaches the maximum lake depth 3 days after becoming ice-free. Additionally, we speculate that these FLake model runs may be biased towards later mixing dates, given that no under-ice mixing appears to occur in the model runs even though under-ice mixing and warming of water beneath lake ice is well documented in Arctic lakes and was also observed at Big Bear Lake.

- Based on these multiple lines of evidence presented above, we think it is sufficient to say that the lakes were well-mixed when they were sampled. We have added the justifications provided above (Big Bear Lake temperature profile, Hille et al. (2009) observations, FLake model runs and a short paragraph explaining the evidence for why lakes were mixed) as an additional appendix to the manuscript.

*Data sources: in accordance with the HESS data policy, please provide a statement of where the stable isotope measurements and meteorological observations used in the study can be accessed.*

- The availability of the isotope data was already provided in the data availability statement; we now provide a link to where the meteorological data can be accessed.

*Specific comments:*

*L13: "limited vertical mixing conditions that lead to this relationship are present at all ice-covered lakes": is this true for all lakes? Exceptions that come to mind include shallow lakes that freeze to the bottom, or are subject to radiatively-driven convection that mixes the water column prior to snowmelt.*

- These lakes are exceptions that we had not considered in this sentence. We have scaled back this sentence to "We expect a similar relationship between increasing lake depth and greater snowmelt bypass could be present at any open-drainage ice-covered lakes that are poorly mixed during the freshet."

*L41-42: "cannot mix with the deeper, warmer, and denser lake waters": but in small lakes, snowmelt can displace these waters, for example to an outflow or to a different basin (Jansen et al., 2019, Cortés & MacIntyre, 2020).*

- We have now added this information to the Introduction

*L54: "up to 25% of the landscape": by surface area? Please specify.*

- Text changed to: "up to 25% of the landscape surface area"

*L56: "isotope methods": what is meant here? Please specify.*

- Text changed to: "We used lake isotope compositions from before and after snowmelt"

*L75: "1981-2010 climate normals": please provide a source for this data.*

- Source added (Environment and Climate Change Canada, 2019)

*L83 and Figure 3: since this information is provided before the methods, for context it would be helpful to list the data source of the meteorological observations.*

- Source added (Environment and Climate Change Canada, 2019)

*L102: how many replicate bottles were collected? How many replicate measurements were performed of each sample?*

- No replicate water samples were collected from lakes, and every fifth sample was analyzed twice. This information was added to the text (shown further down in our responses)

*L102: please specify the container material, as long-term storage in some polyethylene containers can cause isotopic fractionation (Spangenberg, 2012).*

- The text in this line already specified the material as high-density polyethylene (HDPE), which preserve $\delta^{18}O$ and $\delta^{2}H$ values well according to Spangenberg, 2012. "Water samples were collected in 30mL high-density polyethylene bottles"

*L103: what is the manufacturer and model of the laser spectrometer used? How was the instrument calibrated?*

- Text added: "Isotope concentrations were measured using a Los Gatos Research (LGR) Liquid Water Isotope Analyser, model T-LWIA-45-EP at the Environmental Isotope Laboratory at the University of Waterloo. The instrument was calibrated using Vienna Mean Standard Ocean Water (VSMOW) and Vienna Standard Light Antarctic Precipitation (VSLAP) standards provided by LGR. Calibration of the instrument was checked during the analysis using the VSMOW and VSLAP standards."

*L105: this equation appears to be erroneous: the 'x1000' factor is unnecessary when also using the per mil notation. See Coplen, 2011, p. 2554-2555.*

- The variables on the right hand side of the equation (Rsample, RVSMOW) do not use per mil but are ratios of 18O/16O or 2H/1H (L106). These ratios need to be multiplied by 1000 to equate to $\delta$ notation on the left hand side (which we always present in units of ‰). While Coplen (2011) argues that it is confusing to add a multiplier because $\delta$ units are not naturally in ‰ units, we would argue that it is more appropriate to include the 1000x multiplier for this manuscript given that we only present isotope data using ‰ units, and most of the papers we cite also follow this convention. See Equation 2.2 in Kendall & McDonnell, 1998, p. 55

*L108: how were the analytical uncertainties determined?*

- Uncertainties were provided to us by the laboratory, calculated as two standard deviations of the difference between duplicate sample runs according to their website. However, upon further inspection it seems that the values they provided us are default values that are given for every analysis provided by the lab. We calculated the standard deviation using our duplicate sample runs and found the uncertainty to be slightly lower than the laboratory stated. We changed this now and added a description of how we calculated uncertainty.

- Text changed to: "Every fifth sample was analyzed a second time to determine the analytical uncertainties, which were ±0.1‰ for $\delta^{18}O$ and ±0.6‰ for $\delta^2H$, calculated as two standard deviations away from the difference between the duplicate samples."

*L124: "The average difference obtained using the two isotopes in the estimate of the percentage of lake volume replaced by runoff was minimal (1.8%)." Was the difference between isotopes negligible in all lakes? Please provide, in addition to this average number of 1.8%, a minimum and maximum value.*

- Text changed to: "The difference obtained using the two isotopes in the estimate of the percentage of lake volume replaced by runoff was minimal, averaging 1.8% and ranging from 0.6 – 3.7%."

*L131: why were the pre-melt source isotopic signatures ($\delta^{18}O_I$ Ice-Corrected and $\delta^2H_I$ Ice-Corrected in Table 2) also corrected for freezing fractionation? Is this because $\delta_I$ was estimated by drawing a line from $\delta^*$ through $\delta_{L\_corrected}$?*

- The correction for freezing fractionation was applied only to the pre-snowmelt water samples, from which we calculated $\delta_I$. We describe these as "corrected" to clarify that we did not use the uncorrected pre-snowmelt lake water isotope compositions to calculate $\delta_I$. We clarified this in the Table 2 caption.

*L177: "higher isotope composition": unclear what this means.*

- Text changed to: "with a more enriched isotope composition than $\delta_{Snow}$"

*L199: "The presence of a uniformly thick layer of freshet beneath lake likely": rephrase*

- Text changed to "The presence of a somewhat uniformly-thick layer of freshet beneath lake ice likely explains the relationship between lake depth and the amount of lake water replaced by runoff, because the freshet layer represents a relatively smaller portion of lake volume at deeper lakes (Figure 6)."

*L200: Figure 6 does not show that the freshet layer is uniformly thick. To support this claim, it would be helpful to have a (supplementary) table that shows the computed layer thicknesses for each lake.*

- We assume the reviewer is referring to Figure 7. This is a great idea. We never thought about calculating the layer thickness from the % of lake water replaced. We have added a table in the supplementary information, showing that layer thickness (calculated by converting the % of lake water replaced to lake depth by rearranging Equation 5 to solve for $D_{Lake}$;

Equation E1 in the revised version). Layer thickness ranges from 0.12 to 0.52 m, with an average of 0.28 m and standard deviation of 0.11 m (see Table E1). There is no significant relationship between layer thickness and other explanatory variables. We believe these layer thickness values are low compared to other studies that measured layer thickness in situ because they do not account for changes in lake level that are caused by freshet, water which ultimately bypasses the lake.

*L204: "Shallower lakes likely had colder lakebed temperatures": is there a study that shows this is true for thermokarst lakes?*

- We have found a study from the region where they observe that deeper lakes have warmer bed temperatures; text changed to "Shallower lakes likely had colder lakebed temperatures (Burn, 2005)"

*L238: "as minimal hydrological activity occurs over the winter months at arctic lakes" I assume what is meant by "hydrological activity" are water flows from land to lake, but please specify.*

- Yes, this is what we meant. The text now clarifies this: "as there is minimal water flow in and out of arctic lakes during the winter months due to frozen soils and ice cover on lakes (Woo, 1980)."

*L254: In this section I think it may be worthwhile to also briefly discuss the impact of the timing of snowmelt and ice-out on freshet retention and its ecological implications (Dugan, 2021; Hrycik et al., 2021). For example, what happens when the ice goes out before all the snow has melted?*

- We had discussed in the second paragraph briefly about advanced snowmelt timing relative to ice-out timing, which is the likely trajectory of the Arctic. We have now added some discussion about how advanced snowmelt timing may impact lake water chemistry and limnological properties, but we think more research about the composition of freshet runoff, and how it changes over the course of the snowmelt is needed to be able to hypothesize about how these changes will affect lake chemistry and limnology.

*L325: "and calculated for $\delta^{18}O$ as:" shouldn't this read "and calculated for $\delta^2H$ as:"?*

- Text changed to "and calculated for $\delta^2H$ as"

*Figures 1 and 7: the coloured temperature gradient does not show in my preprint copy of the manuscript.*

- The colour gradient was only meant to show in the bar. We understand now how this design is unintuitive (the other reviewer also interpreted the figure in the same way). We have

removed the colour bar and replaced it with a hypothetical graph showing change in lake temperature with depth.

*Figure 5a,b: please define $\delta_{SSL}$. Also, the dashed LEL line from $\delta_*$ to $\delta_P$ appears to curve so as to fit through the point $\delta_{SSL}$. Is this correct?*

- Text added: "$\delta_{SSL}$ is the isotope composition for a terminal basin at steady-state where evaporation equals inflow"
- This is correct, for example see Figure 5c in Tondu et al., 2013.

*Table 1: since snow and ice thickness are variable properties, they seem out of place in a table that lists the lake location and morphometry. Consider moving these two variables to a separate table that includes the sampling date.*

- We have moved these properties from Table 1 to Table 2.

*Table 2: "Pre-Snowmelt Lake Water (01-05-2018)", were all samples collected on this date? Also, for clarity and consistency, consider using $\delta_L$ and $\delta_I$ in the column headers and provide a brief definition in the table caption.*

- Some samples were collected on April 26 and April 30, we changed the text in the table to reflect this now.

References

Burn, C. R.: Lake-bottom thermal regimes, western Arctic Coast, Canada, 16, 355–367, https://doi.org/10.1002/ppp.542, 2005.

Hille, E.: The effects of shoreline retrogressive thaw slumping on the hydrology and geochemistry of small tundra lake catchments, University of Victoria, 189 pp., 2009.

Kendall, C. and McDonnell, J. J.: Isotope Tracers in Catchment Hydrology, Elsevier, Amsterdam, 839 pp., 1998.

Tondu, J. M. E., Turner, K. W., Wolfe, B. B., Hall, R. I., Edwards, T. W. D., and McDonald, I.: Using Water Isotope Tracers to Develop the Hydrological Component of a Long-Term Aquatic Ecosystem Monitoring Program for a Northern Lake-Rich Landscape, Arctic, Antarct. Alp. Res., 45, 594–614, https://doi.org/10.1657/1938-4246-45.4.594, 2013.

Vucic, J. M., Gray, D. K., Cohen, R. S., Syed, M., Murdoch, A. D., and Sharma, S.: Changes in water quality related to permafrost thaw may significantly impact zooplankton in small Arctic lakes, 30, https://doi.org/10.1002/eap.2186, 2020.

**References**

Coplen, T. B.: Guidelines and recommended terms for expression of stable-isotope-ratio and gas-ratio measurement results, Rapid Commun. Mass Spectrom., 25, 2538–2560, https://doi.org/10.1002/rcm.5129, 2011.

Cortés, A. and MacIntyre, S.: Mixing processes in small arctic lakes during spring, Limnol. Oceanogr., 65, 260–288, https://doi.org/10.1002/lno.11296, 2020.

Dugan, H. A.: A Comparison of Ecological Memory of Lake Ice†Off in Eight North†Temperate Lakes, J. Geophys. Res. Biogeosciences, 126, https://doi.org/10.1029/2020JG006232, 2021.

Hrycik, A. R., Isles, P. D. F., Adrian, R., Albright, M., Bacon, L. C., Berger, S. A., Bhattacharya, R., Grossart, H., Hejzlar, J., Hetherington, A. L., Knoll, L. B., Laas, A., McDonald, C. P., Merrell, K., Nejstgaard, J. C., Nelson, K., Nõges, P., Paterson, A. M., Pilla, R. M., Robertson, D. M., Rudstam, L. G., Rusak, J. A., Sadro, S., Silow, E. A., Stockwell, J. D., Yao, H., Yokota, K., and Pierson, D. C.: Earlier winter/spring runoff and snowmelt during warmer winters lead to lower summer chlorophyll†a in north temperate lakes, Glob. Chang. Biol., 27, 4615–4629, https://doi.org/10.1111/gcb.15797, 2021.

Jansen, J., Thornton, B. F., Jammet, M. M., Wik, M., Cortés, A., Friborg, T., MacIntyre, S., and Crill, P. M.: Climate-sensitive controls on large spring emissions of CH4 and CO2 from northern lakes, J. Geophys. Res. Biogeosciences, 124, 2379–2399, https://doi.org/10.1029/2019JG005094, 2019.

Spangenberg, J. E.: Caution on the storage of waters and aqueous solutions in plastic containers for hydrogen and oxygen stable isotope analysis, Rapid Commun. Mass Spectrom., 26, 2627–2636, https://doi.org/10.1002/rcm.6386, 2012.

Vachon, D., Langenegger, T., Donis, D., and McGinnis, D. F.: Influence of water column stratification and mixing patterns on the fate of methane produced in deep sediments of a small eutrophic lake, Limnol. Oceanogr., lno.11172, https://doi.org/10.1002/lno.11172, 2019.

*Wiltse, B., Yerger, E. C., and Laxson, C. L.: A reduction in spring mixing due to road salt runoff entering Mirror Lake (Lake Placid, NY), Lake Reserv. Manag., 36, 109–121, https://doi.org/10.1080/10402381.2019.1675826, 2020.*

Reviewer 2

*This is an interesting manuscript that presents a novel way to estimate the amount of lake water that is replaced by freshet near the end of winter by using isotope analysis. The authors also linked the amount of water replaced by freshet with lake depth. Furthermore, the isotope analysis suggested that the freshet have different sources such as snow or rainwater from the previous fall. I think the results are publishable, and just have a few minor comments about the manuscript that hopefully will help improve it.*

Thank you for your thorough review of the manuscript which has helped improve its quality.

1. *I suggest a slight revision of the title. The authors can delete "Northwest territories, Canada" without loss of clarity. The paper also goes into detail on how the different lake/watershed parameters influence the % of lake water replaced by freshet, but it is not clear from the title.*

   • Thank you for your suggestions. We have changed the title to incorporate them and it makes the title more impactful we believe. The title now reads "Assessing the influence of lake and watershed attributes on snowmelt bypass at thermokarst lakes"

2. *The authors can expand on why they chose to look at watershed and lake characteristics and how that affected the magnitude of snowmelt bypass in the introduction. Did any previous studies or preliminary data hint on this? The authors could also expand on the potential impacts of the study on the ecology in the conclusion.*

   • We have now expanded on why we thought lake depth or watershed size may impact snowmelt bypass, due to the way freshet water mixes under lake ice, and observations of how freshet water can displace pre-snowmelt lake water if there is enough volume. We also add that knowledge of how lake and watershed attributes affect snowmelt bypass allows us to better interpret and understand why lake water chemistry and limnological properties may vary from lake to lake in a given region, and expand in the conclusion about the implications of our findings on lake ecology.

3. *In figure 7 the schematic ends with the lake being well mixed throughout the water column. However, the authors did not show that the lakes were well mixed post snowmelt. Some arctic lakes (e.g Cortes and Macintyre 2020) experience mixing only in the upper part of the water column at the end of winter and spring. This well-mixed condition was also implicitly assumed in equation 2 when the authors estimate the % of lake water replaced. I realize that these lakes are significantly shallower than the one presented in Cortes and Macintyre (2020), but the formula might need to be adjusted in the case of incomplete mixing. If this is possible, then the methods in this paper could be extended to a wider class of lakes.*

   • The assumption that lakes were well mixed post-freshet is an assumption that we did overlook in this manuscript. We now provide a few different lines of evidence in an

additional manuscript appendix that all point towards lakes being well-mixed when they became ice free. It would also be interesting for future studies to investigate snowmelt bypass variability at deeper lakes that do not fully mix, which we now mention in the Discussion.

- In the new appendix we provide water temperature data from Big Bear Lake at 1.25m, showing that the lake water temperature at this depth reaches 4°C initially by 05-06-2018, followed by daily fluctuations between 2.3 – 4.1°C, before continuing a warming trend again on 13-06-2018 and reaching 6.8°C on 15-06-2018.

- Additionally, we found a lake water temperature profile measured by Hille (2009) at a 10.9 m deep lake 10 km of the Inuvik-Tuktoyaktuk highway. Hille's data (Figure 3.4) show uniform warming of the water temperature between 1 and 4 metres from ~2°C to ~5°C by the time the lake became ice-free.

- As suggested by the other reviewer, we also ran "FLake-Global: online modelling system" (http://flake.igb-berlin.de/model/run) using two different sets of parameters representing a "typical" lake from our dataset, and for a small/deep lake where we would expect the least favourable conditions for mixing after lakes became ice-free. We opted for a "perpetual year" run where meteorological data from 01-10-2015 to 30-09-2016 is used to force the model for multiple years until a quasi-stationary state is reached. The model parameters for the 'typical' lake is based roughly on averages taken from Table 1 in the manuscript: lake fetch was set at 250 m (a perfectly circular lake of this fetch would be 5 ha, close to the median lake size of 4.8 ha), mean lake depth was set to 2 m, and the coordinates used were the average coordinates of the lakes we sampled. The water type was "clear (2m transparency) based on an average Secchi depth of 1.88 m measured at lakes along the Inuvik-Tuktoyaktuk Highway and Dempster highway south of Inuvik by Vucic et al. (2020). This model run simulates the mixed layer depth reaching the maximum lake depth within one day of the lake becoming ice-free. We ran the second "worst chance of mixing" scenario with a mean depth of 3.5 m, and a fetch of 50 m, with other parameters identical to the first scenario. In this scenario, the mixed layer reaches the maximum lake depth 2 days after becoming ice-free. Additionally, we speculate that these FLake model runs may be biased towards later mixing dates, given that no under-ice mixing appears to occur in the model runs even though under-ice mixing and warming of water beneath lake ice is well documented in Arctic lakes and was also observed at Big Bear Lake.

4. *Line 14: If the lake freezes all the way to the bottom in winter, then this condition does not exist.*

- This is true. We had overlooked this class of lakes. We have changed the text to: "We expect a similar relationship between increasing lake depth and greater snowmelt bypass could be present at any open-drainage ice-covered lakes that are poorly mixed during the freshet."

5. *There is a colourbar for the schematic in figure 1 but I do not see the colours. It could be worthwhile to put the same colourbar in figure 7. In figure 7 the authors can label what the brown arrows are, as figure 1 did.*

- The colour gradient was only meant to show in the bar, we understand now how this design is unintuitive (the other reviewer also interpreted the figure in the same way). We have removed the colour bar and replaced it with a hypothetical graph showing change in lake temperature with depth for both figures. We also added labels to Figure 7 as suggested.

6. *The authors should define in Figure 5*

- We have added definitions for $\delta^*$ and $\delta_{SSL}$ to the figure caption.

7. *Line 200: Not sure how the presence of a layer of freshet explains the relationship between lake depth and the amount of water replaced by runoff, please explain.*

- We have reworded this to: "The presence of a somewhat uniformly thick layer of freshet beneath lake ice likely explains the relationship between lake depth and the amount of lake water replaced by runoff, because the freshet layer represents a relatively smaller portion of lake volume at deeper lakes (Figure 6)."

8. *Line 204: "Shallower lakes likely had colder lakebed temperatures". Does this only apply to thermokarst lakes? In some mid-latitude seasonally ice-covered lakes the bottom can be very close to 4oC because of heat stored in the shallow sediments over summer that flows down via gravity currents. Figure 7a should also be modified.*

- We have clarified that we mean shallower lakebed temperatures during the freshet. We also provide a citation from a study in this region (Burn, 2005) which observed that shallower lakes had cooler lakebed temperatures.

9. *Line 249: Authors should explain what the typical thermal structure is. A recent analysis by Yang et al. (2021) suggests that there can be many typical thermal structures across different ice-covered lakes.*

- We believe these are cryomictic lakes, per Yang et al. (2021), given that these are relatively small lakes, and our FLake model runs show a similar temperature profile to the typical cryostratified lake outlined in Figure 1a of Yang et al. (2021). Notably, the model runs indicate that no surface restratification occurs leading up to ice formation, and the whole lake water column is well mixed until reaching near-zero temperatures before ice formation begins. We clarify in the text now that these are likely cryomictic lakes.

10. *Line 465: Not sure if a software needs to be cited here.*

- We have cited the software used to make our results as easily reproducible as possible.

11. *Table 1: Is the snow depth uniform across the lakes?*

- Snow depth tends to be very uniform across the lakes, with the exception of deeper snow 5-10 metres around the edges of the lake depending on the surrounding topography. We have added this information to the text in the Methods section when describing the collection of snow depth data.

12. *Table 3: The caption says that the p-values are shown for each isotope but there is only 1 value.*

- This is an error. We had originally used both isotopes however we decided to use just one since the values they gave were nearly identical. We have removed this part.

13. *Table 3: The authors can reorder the variables such that the lake parameters come first, then the other variables after.*

- We have reordered the table as you suggest.

14. *HESS requires a data availability section at the end of the manuscript on how the data used in this paper can be accessed.*

- Our data availability statement now provides a link to downloading the isotope data, lake and watershed characteristics, and meteorological data used in the study.

15. *I suggest the authors change the notation for the fraction of total lake volume, as VLake looks a lot like the volume of the lake.*

- We have changed the notation from "$V_{Lake}$" to $\%_{LakeVolume}$

References:

Burn, C. R.: Lake-bottom thermal regimes, western Arctic Coast, Canada, 16, 355–367, https://doi.org/10.1002/ppp.542, 2005.

Cortés, A., & MacIntyre, S. (2020). Mixing processes in small arctic lakes during spring. Limnology and Oceanography, 65(2), 260-288.

Yang, B., Wells, M. G., McMeans, B. C., Dugan, H. A., Rusak, J. A., Weyhenmeyer, G. A., ... & Young, J. D. (2021). A new thermal categorization of ice― covered lakes. Geophysical Research Letters, 48(3), e2020GL091374.

---

## Author Response (AR2)

Author responses

RV: L105: this equation appears to be erroneous: the 'x1000' factor is unnecessary when also using the per mil notation. See Coplen, 2011, p. 2554-2555.

AUTH: The variables on the right hand side of the equation (Rsample, RVSMOW) do not use per mil but are ratios of 18O/16O or 2H/1H (L106). These ratios need to be multiplied by 1000 to equate to δ notation on the left hand side (which we always present in units of ‰). While Coplen (2011) argues that it is confusing to add a multiplier because δ units are not naturally in ‰ units, we would argue that it is more appropriate to include the 1000x multiplier for this manuscript given that we only present isotope data using ‰ units, and most of the papers we cite also follow this convention. See Equation 2.2 in Kendall & McDonnell, 1998, p. 55

RV: Per mil notation works the same way as percentage notation: 46/100 = 0.46 = 46%. There is no 100x multiplier. Similarly, for the per mil notation no multiplier is necessary. Delta is dimensionless ratio, so there are no units that need correcting.

- We have removed the 1000x multiplier from the equation.

Minor comments:

L5. "hydrological and limnological": I'm not sure how one would define "limnological properties of the lake" – maybe stick with "hydrological and biogeochemical".

- Suggested change implemented.

L11: "The thickness of the freshet layer was not proportional to lake depth, which isolated a larger portion of pre-snowmelt lake water from mixing at deeper lakes." -> The thickness of the freshet layer was not proportional to maximum lake depth, so that a relatively larger portion of pre-snowmelt lake water remained isolated in deeper lakes.

- Suggested change implemented.

L15: poorly mixed -> partially mixed

- Suggested change implemented.

L41: "the flow of runoff into lakes", it would be helpful to have a more quantitative definition of 'freshet'. E.g. the volume of meltwater entering the lake. In the current version of the MS it seems that 'freshet' is used to describe both the volume of meltwater flowing into the lake, and the time (event) when this happens.

- "" → "the volume of snowmelt-driven runoff flowing into lakes"

L66: "lake water isotope composition" -> isotopic composition of lake water and precipitation

- Suggested change implemented.

L72: "inform assessments of hydrological and limnological properties of thermokarst lakes". What does this mean? I think it would be possible to be more specific here. This study describes which lakes retain meltwater and what the source of the meltwater is, which are important and novel results. But why do we want to know this? E.g. if more freshet is retained (e.g. because more lakes become closed basins), what does this mean for (annual) cycles of carbon, nutrients and energy in thermokarst lakes? Does it matter whether meltwater originates from snow or rain?

- We have reworded this sentence to specify how knowledge of how lake and watershed attributes influence snowmelt bypass will help others in their assessments of hydrological and limnological properties of lakes.
- "" → "Future assessments of hydrological and biogeochemical properties of thermokarst lakes can use the lake and watershed attributes we identify to affect snowmelt bypass and lake water sources to inform their results, given the distinct biogeochemical properties of freshet runoff (Finlay et al., 2006, Balasubramaniam et al., 2015) and the influence of snowmelt bypass on the amount of freshet runoff retained by lakes." Sentence replaced.

L103: it might be good to mention that you are measuring stable isotopes only, to distinguish from radiotracer studies.

- "" → "for stable isotope analysis"

L110 and elsewhere: isotope composition -> isotopic composition (of water), or oxygen and hydrogen isotope composition

- "" → "isotopic composition", throughout manuscript.

L124: please see my response above about per mil notation

- We have removed the 1000x multiplier from the equation.

L143: I don't understand this sentence. In other years the lakes were not exposed to meteorological conditions (weather?) during the open water season?

- The most recent meteorological conditions are used because they are what will have had the largest impact on the isotopic composition of the lake water. We did not use 2018 data because the lakes were ice-covered during the sampling campaign, barring the last few days (i.e., not an entire open-water season).

L155: paramtr -> parameter

- Suggested change implemented.

L247: "Shallower lakes likely had colder lakebed temperatures during freshet (Burn, 2005), which allowed more mixing between pre-snowmelt lake water and freshet inflow due to the reduction in

water density gradient between the bottom of the lake and the top of the lake." Wouldn't this result in a deeper mixing layer in smaller lakes, rather than the uniform thickness estimated here?

- This is true, but based on our results it seems that the colder lakebed temps at shallower lakes had a relatively small impact on total freshet retention when compared to lake depth. We have clarified this now.
- "hallower lakes likely had colder lakebed temperatures during freshet (Burn, 2005),  more mixing between pre-snowmelt lake water and freshet inflow due to the reduction in water density gradient between the bottom of the lake and the top of the lake." → "We hypothesized that because shallower lakes likely had colder lakebed temperatures during freshet (Burn, 2005), more mixing between pre-snowmelt lake water and freshet inflow would occur due to the reduction in water density gradient between the bottom of the lake and the top of the lake. However, the estimated thickness of the freshet layer was uniform across lakes, indicating that colder lakebed temperatures may not have contributed to greater mixing at shallower lakes." Added parts are underlined.

L255: the subarctic lakes in the study of Jansen et al. (2019) and the arctic lakes in Cortés and MacIntyre (2020) are within the size range of the lakes in this study. Could it be that the volume and loading of snowmelt also determines the fraction of lake water replaced by meltwater? Another aspect to consider is that chemical gradients in smaller lakes are often stronger under ice (compared to larger lakes), which increases density stratification and limits meltwater mixing and retention (Cortés & MacIntyre, 2020).

- We have deduced that the volume of freshet runoff does not impact the fraction of lake water replaced by freshet runoff because there was no relationship between relative watershed area (watershed area/lake area) and the fraction of lake water replaced by freshet runoff.
- We have added the bit about stronger chemical gradients in smaller lakes, which seems to suggest they can experience some bypass if they are more heavily stratified.
- "However, smaller lakes <1 ha, which are common in the Arctic (Pointner et al., 2019), likely do not experience snowmelt bypass because freshet is able to displace the pre-snowmelt lake water due to the small volume of the lake (Jansen et al., 2019, Cortes and MacIntyre, 2020)." → "However, smaller lakes <1 ha, which are common in the Arctic (Pointner et al., 2019), likely do not experience as strong a snowmelt bypass effect because freshet is able to displace the pre-snowmelt lake water due to the small volume of the lake (Jansen et al., 2019, Cortes and MacIntyre, 2020). However, smaller lakes also typically have smaller under-ice chemical gradients that increase density stratification and limit freshet mixing and retention (Cortes and MacIntyre, 2020)." Added parts are underlined.

L341: One might add that snowmelt in spring can constitute the largest annual influx of terrestrial organic matter to lakes in organic-rich landscapes (e.g. Townsend-Small et al., 2011, https://doi.org/10.1007/s10533-010-9451-4 and Olefeldt & Roulet, 2012 https://doi.org/10.1029/2011JG001819), and so the retention of meltwater could affect the NEP and ultimately climate feedbacks of these lakes.

- "Snowmelt also tends to have higher dissolved organic carbon (DOC) concentrations than summertime runoff (Finlay et al., 2006) and typically contributes the largest input of terrestrial organic mater to lakes in organic-rich landscapes (Townsend-Small et al., 2011;

Olefeldt and Roulet, 2012). Balasubramaniam et al. (2015) observed that thermokarst lakes dominated by snow-sourced water tended to have lower pH, higher conductivity and higher DOC concentrations than lakes dominated by rain-sourced water. Based on these observations, as snowmelt occurs earlier in the Arctic, lakes may experience decreases in DOC, and conductivity, and increases in pH. Such changes to lake biogeochemistry caused by shifts in freshet runoff retention could affect the productivity and ultimately the climate feedbacks of these lakes." Added parts are underlined.

L367: "likely to bypass" -> suggest nuancing this statement, since some meltwater is retained.

- " large volume of freshet that flows into lakes every year  bypass ice-covered, open-drainage lakes due to limited mixing between lake water beneath the lake ice and freshet" → "A portion of the large volume of freshet that flows into lakes every year can bypass ice-covered, open-drainage lakes due to limited mixing between lake water beneath the lake ice and freshet." Added parts are underlined.

L370: "because freshet is unable to mix with deeper lake water": the authors make a strong case for this hypothesis but lack the conductivity and temperature observations to demonstrate the extent of mixing. I would suggest nuancing this statement a bit.

- "Our data show that as lake depth increases the amount of lake water replaced by freshet decreases, because freshet is unable to mix with deeper lake water when lakes are ice-covered and the water column is stratified." → "Our data show that as lake depth increases the amount of lake water replaced by freshet decreases, likely because freshet is unable to mix with deeper lake water when lakes are ice-covered and the water column is stratified, however we lack data demonstrating the extent of mixing in the lakes we studied." Added parts are underlined.

L371: "Additionally, the volume of freshet flowing into the lakes seems to have no observable impact on the amount of lake water replaced by freshet." How was this conclusion arrived at? Was the volume of freshet estimated somewhere?

- We assume this based on the relative size of a lake's watershed (watershed area/lake area) having no impact on the % of the lake water replaced by freshet. There were some lakes that had watersheds 17 times the size of the lake, while other lakes had watersheds only 1.9 times the size of the lake (Table 1). We assume that relative watershed size is a good proxy for the volume of freshet inflow lakes received. We now specify how we reach this conclusion.
- "Additionally, the volume of freshet flowing into the lakes seems to have  impact on the amount of lake water replaced by freshet." → "Additionally, the volume of freshet flowing into the lakes seems to have minimal impact on the amount of lake water replaced by freshet, given that the ratio of watershed area to lake area was not correlated with the percentage of lake water replaced by freshet." Added parts are underlined.